# Effect of High-Temperature Calcined Wheat Straw Powder after Lignin Removal on Properties of Waterborne Wood Coatings

**Xiaoxing Yan [1,2,*], Lin Wang [2] and Xingyu Qian [2]**

[1] Co-Innovation Center of Efficient Processing and Utilization of Forest Resources, Nanjing Forestry University, Nanjing 210037, China

[2] College of Furnishings and Industrial Design, Nanjing Forestry University, Nanjing 210037, China

* Correspondence: yanxiaoxing@nuaa.edu.cn; Tel.: +86-25-8542-7528

**Abstract:** The effect of adding wheat straw powder after lignin removal (WSPALR) and high-temperature calcined WSPALR on the hardness, adhesion, and resistance to impact, color difference, and mold resistance of waterborne coatings was studied. The results showed that the hardness was the highest of 6H when the concentration of WSPALR was 1.0%–2.0%. WSPALR and high-temperature calcined WSPALR had little effect on the adhesion and impact resistance of waterborne coatings, and the resistance to impact was about 10.0 kg cm. When both the concentration of WSPALR and high-temperature calcined WSPALR were 0.5%, the waterborne coating had the best adhesion of Level 1. The addition of high-temperature calcined WSPALR maintained the color difference of the original coatings. A high WSPALR concentration showed better mold resistance than a low concentration WSPALR, and the inhibition effect of high-temperature calcined WSPALR on Trichoderma was better than that of WSPALR. When the concentration of WSPALR calcined at a high temperature was 0.5%, it showed a better hardness of 4H, Level 1 adhesion, 10.0 kg cm resistance to impact, and 1.1 color difference of the waterborne coating. This work has important application value for mold resistance of wood coatings.

**Keywords:** waterborne coatings; wheat straw powder after lignin removal (WSPALR); calcined WSPALR; mechanical properties; resistance to molds

## 1. Introduction

In recent years, people's awareness of health and environmental protection has gradually increased, and the relevant environmental protection policies and regulations have been further improved [1]. Waterborne coatings, powder coatings, high solid coatings, and other new coatings have become a new direction in the development of coatings [2]. Among them, waterborne coatings use waterborne or water-emulsion polymer materials as binders or main film-forming materials, as well as cellulose or polyacrylate as thickeners [3]. Compared with traditional coatings, waterborne coatings have the advantages of rational use of resources, non-pollution of the environment, and less harm to the human body [4]. The most important feature of waterborne coatings is that they do not contain volatile organic compounds and air pollutants [5]. At present, waterborne coatings have become the focus of research and development in the field of coatings [6]. Wood is rich in various nutrients such as sugar and starch for microbial growth, and therefore it is not resistant to fungal and bacterial infestations, which reduce the wood quality and use value and make it not conducive to subsequent processing and utilization [7,8]. When waterborne coatings are applied to the wood surface, mold may cause discoloration and even decrease the adhesion strength of the film, which will affect the service life of products, and bring serious economic losses to manufacturers and users [9].

At present, many researchers have done much work towards improving the water resistance, corrosion resistance, and mechanical properties of waterborne coatings [10]. Xu et al. [11], Lu et al. [12], and Li et al. [13] prepared waterborne polyurethane and crosslinked polyurethane-acrylate composites to improve the mechanical properties of the waterborne wood coatings. Lin et al. [14], Balakrishnan et al. [15], and Zhao et al. [16] synthesized a novel waterborne polyurethane to enhance the water resistance and thermostability. Jiang et al. [17], Kozlova et al. [18], Verma et al. [19], and Chen et al. [20] enhanced the anti-corrosion behaviors of waterborne coatings. However, there have been few studies on improving the mold resistance of waterborne wood coatings. Therefore, it is expedient to improve the performance of waterborne coatings so as to improve the mold resistance of wood.

In order to pursue the concept of green environmental protection [21], first of all, we should consider the green environmental protection performance of mold-resistant substances added to waterborne coatings. However, most existing mold-resistant additives on the market are mainly chemically synthesized organic compounds, which have a certain toxicity and cause harm to the human body [22]. Therefore, using natural plant and marine biological extracts to prepare natural mold-resistant substances instead of the chemical synthetic mold-resistant products currently used in the market is of great significance to improve the safety and environmental protection performance of wood products based on waterborne coatings. Cellulose nanocrystals, as the largest and most widely distributed natural polymer in nature, are a valuable renewable resource that has not been fully utilized by human beings [23]. Cellulose nanocrystals not only have the characteristics of nanoparticles but also have some unique strength and optical properties [24]. Cellulose nanocrystals have the advantages of high crystallinity, a high Young's modulus of elasticity, high physical strength, large specific surface area, lightweight, biodegradability, biocompatibility, and regeneration of biomaterials, showing its unique potential and broad application prospects [25]. However, cellulose nanocrystals are not easy to prepare and obtain, and they are very expensive. If filamentous substances can be obtained by a simple method and have good mold resistance, they will have good prospects. Wheat straw is the residual part of wheat left after harvesting the seeds. It is a kind of renewable biological resource with many uses, high fiber content, and lignin. The reason for using the calcined cellulose is that wheat straw is rich in potassium, calcium, magnesium, and organic matter. Carbon is converted into carbon dioxide after calcination at a high temperature, which volatilizes and forms micropores. The micropores are active and can prevent fungi from multiplying. Therefore, it is possible to improve the mold resistance using the calcined cellulose [26].

In this paper, wheat straw powder after lignin removal (WSPALR) was obtained from wheat straw treated with alkaline hydrogen peroxide. The effect of WSPALR and high-temperature calcined WSPALR on the mold resistance of waterborne coatings was studied, and the color difference, hardness, adhesion, and resistance to impact of waterborne coatings were also studied, from which the properties of waterborne coatings were analyzed to improve the mold resistance of waterborne coatings while maintaining their color difference, hardness, adhesion, and resistance to impact properties and elongation at break.

## 2. Materials and Methods

### 2.1. Test Materials

Wheat straw powder, waterborne wood coating, putty (talc powder 80.0 g, water 200.0 g, mixed evenly), poplar veneer (Chinese White Poplar, uniform material color, 100 mm × 100 mm × 5 mm, 300 pieces, after ordinary mechanical sanding) were supplied by Yihua Lifestyle Technology Co., Ltd., Shantou, China. The waterborne wood coating consisted of acrylic copolymers supported by water (the concentration was 90.0%), dipropylene glycol methyl ether (the concentration was 2.0%), and dipropylene glycol butyl ether (the content was 8.0%). The solid concentration of the coating was about 26.5%. The capacity of the Ford cup No. 4 was 100 mL. When the container was full

of the waterborne wood coating, the time it took to flow out from the standard bottom hole was used to measure the coating viscosity in seconds (s) [27]. The viscosity of the waterborne wood coating was 58 Pa s. Hydrogen peroxide solution ($M_W$: 34.01 g/mol, CAS No.: 7722-84-1) and NaOH ($M_W$: 40.00 g/mol, CAS No.: 1310-73-2) were supplied by Daguangming Chemical Reagent Nanjing Co., Ltd., Nanjing, China. All reagents in the experiment were used without further processing.

## 2.2. Preparation of Coatings

The lignin in wheat straw powder was separated by the alkaline hydrogen peroxide method. The wheat straw powder was sieved through 250 μm mesh. According to the calculated solid mass-liquid volume ratio of 1:20, 15.0 g wheat straw powder after sieving was added to 300 mL of the hydrogen peroxide solution. Then, the pH value of the solution was adjusted to 11.5 with NaOH. The solution was poured into a beaker and placed in a collector-type constant temperature magnetic water bath. The reaction time was 24 h at 70 °C. After the reaction, the mixture was centrifuged for 10 min at a speed of 6000 rpm in a high-speed centrifuge and then cleaned with distilled water, repeated several times until the wash water was neutral. The residue was freeze-dried to obtain the dried powder, namely wheat straw powder after lignin removal (WSPALR).

Calcination at 920 °C was selected for the full volatilization of carbon and hydrogen in WSPALR, which can help improve the properties of coatings [28]. The WSPALR was placed in a clean ceramic crucible and the cup covered. The samples were then placed in a high-temperature calciner at a heating rate of 5 °C/min, heated to 920 °C and kept for 45 min to obtain high-temperature calcined WSPALR. The weight loss of wheat straw powder after hydrogen peroxide and sodium hydroxide treatment was 47.0%, and that of WSPALR after high-temperature calcination was 35.4%. The tap bulk density of wheat straw powder, WSPALR and WSPALR after high-temperature calcination was $200.0 \pm 5.0$ kg/m$^3$, $300.0 \pm 5.0$ kg/m$^3$ and $800.0 \pm 5.0$ kg/m$^3$, respectively.

The main ingredients of putty are talc powder and water, and the role of putty was to fill the vessels of poplar wood. Talc powder (80.0 g) was added to 200.0 g water and mixed evenly to obtain the putty. Then the putty was coated evenly on the poplar as a veneer. When the putty was dry, the poplar veneer was sanded using 600 grit sandpaper, and a dry cloth was used to wipe off the dust. The waterborne wood coating was coated using a SZQ tetrahedral fabricator (Tianjin Jinghai Science and Technology Testing Machinery Factory, Tianjin, China) on poplar veneers. The veneer was fixed on the platform. Then, the prepared coatings were poured in front of the fabricator. The two ends of the fabricator were grasped by hand, then glided at a uniform speed of 150 mm/s, and the required thickness of the coating can be coated. After 30 min for natural drying, the coating was sanded using 1000 grit sandpaper; then a dry cloth was used to wipe off the dust. The preparation process of waterborne wood coating needed to be repeated twice. WSPALR (0.5 g) was added to 99.5 g waterborne wood coatings to form 0.5% concentration of WSPALR and mixed evenly. The waterborne coatings with 1.0%, 2.0%, 3.5%, and 5.0% concentrations of WSPALR were also prepared. The formulations are summarized in Table 1. The waterborne coatings with different concentrations of WSPALR were prepared separately using the SZQ tetrahedral fabricator. The coating was naturally dried for 3 h and then sanded using 1000 grit sandpaper, and a dry cloth was used to wipe off the dust. The above process was repeated twice. Similarly, waterborne coatings with 0, 0.5%, 1.0%, 2.0%, 3.5%, and 5.0% concentrations of calcined WSPALR were also prepared (Table 1). The preparation method of waterborne coating with the addition of WSPALR after calcination at a high temperature was the same as that described above. The coating was also naturally dried for 3 h and then sanded using 1000 grit sandpaper, and a dry cloth was used to wipe off the dust. The thickness of the dry waterborne coating was about 40 μm. The waterborne coatings without WSPALR and high-temperature calcined WSPALR were prepared according to the same process and the coating thickness was about 40 μm for comparison. The waterborne coatings with and without WSPALR and high-temperature calcined WSPALR were prepared onto glass substrates according to the same process and the coating thickness was about 40 μm for tensile test measurement.

**Table 1.** Composition of waterborne coatings.

| Sample | Concentration (%) | WSPALR (g) | Calcined WSPALR (g) | Waterborne Wood Coating (g) |
|--------|-------------------|------------|---------------------|------------------------------|
| 1 | 0 | 0 | 0 | 100.0 |
| 2 | 0.5 | 0.5 | 0 | 99.5 |
| 3 | 1.0 | 1.0 | 0 | 99.0 |
| 4 | 2.0 | 2.0 | 0 | 98.0 |
| 5 | 3.5 | 3.5 | 0 | 96.5 |
| 6 | 5.0 | 5.0 | 0 | 95.0 |
| 7 | 0.5 | 0 | 0.5 | 99.5 |
| 8 | 1.0 | 0 | 1.0 | 99.0 |
| 9 | 2.0 | 0 | 2.0 | 98.0 |
| 10 | 3.5 | 0 | 3.5 | 96.5 |
| 11 | 5.0 | 0 | 5.0 | 95.0 |

*2.3. Performance Test*

The microstructures of wheat straw powder, WSPALR, and high-temperature calcined WSPALR were analyzed using a Quanta 200 environment scanning electron microscope (SEM) with EDX (FEI Company, Hillsboro, OR, USA). The samples adhered to the sample pedestal, and the appropriate time and voltage of gold plating were set for gold plating. After the gold plating was finished, the samples were taken out and placed in the sample chamber. When the vacuum degree reached a certain value, the pressure was increased and the SEM of the samples was observed. The coatings were placed in a high-temperature calciner at a heating rate of 5 °C/min, heated to 920 °C and kept for 45 min to obtain high-temperature calcined coating for EDX analysis to calculate the Si concentration. The composition of wheat straw powder and WSPALR were analyzed by a VERTEX 80V infrared spectrum analyzer (Germany Bruker Co., Ltd., Karlsruhe, Germany). Wheat straw powder and WSPALR were observed by pressing method. One to two milligram of powder and 200 mg pure KBr were mixed evenly and placed in the mold. The sample was pressed into transparent tablets for testing. The coatings on the wood surface were made using the attenuated total refraction (ATR) method. The samples were placed above the ATR accessory to make the samples contact closely with the ATR crystal. The infrared beam attenuated and reflected in the ATR crystal (diamond) and reached the detector. The HP-2136 color difference meter (Zhuhai Tianchuang Instrument Co., Ltd., Zhuhai, China) was used to directly measure the lab values of coatings. According to the standard GB/T1732-93 [29], the resistance to impact was measured by the QCJ impactor (Tianjin Jingkelian Material Testing Machine Co., Ltd., Tianjin, China). In the impact test, a ball was dropped on the surface of the coating (the maximum distance between the ball and the coating was 50.0 cm; the weight of the ball was 1.0 kg). At the end of the impact test, the deformation of the coating was observed. If the coating did not crack, the maximum drop height of the ball was the resistance to impact value. According to GB/T 1720-89 [30], a QFZ-II circle-cut coating adhesive testing machine (Tianjin JingKelian Material Testing Machine Co., Ltd.) was used to determine the adhesion of the coating. The experiment was carried out at 23 ± 2 °C and 50% ± 5% relative humidity. When measuring, the tip of the rotary needle touched the coating, and the handle was shaken uniformly clockwise. The rotational speed was in the range of 80–100 r/min, and the scratch length of the circular roller was 7.5 ± 0.5 mm. At the end of the adhesion test, the damage to the coating was observed using a magnifier. The marks 0, 1, 2, 3, 4, 5, 6, and 7 on the top of the coating were the eight levels of damage. The adhesion of level 0 was the strongest, and that of Level 7 was the weakest. The hardness of the coating was tested with a 6H–6B pencil. In this experiment, the angle between the pencil and the coating was 45°, and the pencil scratched under a 1.0 kg load. The hardness of the film (determined with 6H, 5H, 4H, 3H, 2H, 1H, HB, 1B, 2B, 3B, 4B, 5B, and 6B pencils) was measured when scratches appeared on the coatings. The digital microscopic images of wheat straw powder, WSPALR, and high-temperature calcined WSPALR were observed using a Zeiss AXIO Scope A1 microscope (Carl Zeiss AG, Oberkochen, Germany). The elongation at break of the coatings was tested by the Model AG-IC100KN precision electronic universal capability experiment machine (Shimadzu Co., Ltd., Kyoto, Japan) and TRview X optical displacement meter (Shimadzu Co.,

Ltd., Kyoto, Japan). The coating was peeled off the glass substrate and the elongation at break of the coating was calculated according to the displacement length of the coating at break and the original length of the coating before stretching. The coatings were prepared into thin strips, and then the two ends of the coatings were fixed with fixtures. At a certain tension speed (0.12 mm/min), the coating was deformed until it was destroyed by applying a load on the longitudinal direction of the coatings. During the tensile test, the two ends of the coating sample were fixed firmly to ensure that the coating did not slide. In the tensile test, the crack was not in the coating chuck, but in the middle of the coating. Mold resistance was tested by the suspension method according to GB/T 1741-2007 [31]. The samples were placed in an incubator. The bottom of the incubator had enough exposed water surface area to ensure that the sample was not touched or splashed by water and that the samples were not placed in contact with each other. The temperature was between 25 and 30 °C and the relative humidity was 85.0%. After 5 days, mold growth on the surface was observed. The waterborne coatings can make the wood expansion rate less than 0.5%. The coated wood samples hardly swelled and the swelling could be neglected in the mold resistance test. All the experiments were repeated at least four times with an error of less than 5.0%.

## 3. Results and Discussion

The SEM image and elemental analysis of wheat straw powder, WSPALR, and high-temperature calcined WSPALR are shown in Figure 1. The SEM image of wheat straw powder showed a layered structure (as shown in the red box in Figure 1A). EDX analysis showed that it contained a large amount of carbon and silicon. After treatment with hydrogen peroxide and sodium hydroxide, it was long rods, about 200 μm in length and 5 μm in diameter. Elemental analysis showed that there was still a large amount of carbon and silicon. WSPALR was long, rod-like, and accompanied by some tubular substances (as shown in the red box in Figure 1C). It showed that treatment with hydrogen peroxide and sodium hydroxide turned it into micron filamentous products, which were cellulose and $SiO_2$ (Si comes from ash in wheat straw powder). After calcination at a high temperature, carbon was basically removed as a granular material. The component analysis showed that the material was basically carbon-free, and most of it was silicon. This showed that carbon was converted into carbon dioxide and evaporated after calcination at a high temperature, leaving many micropores, which were porous and irregular composite materials (as shown in the red box in Figure 1E). The SEM image and EDX analysis of the waterborne coatings with WSPALR and calcined WSPALR added are shown in Figure 2, indicating that a small amount of agglomeration of WSPALR or calcined WSPALR occurs due to the large numbers of particles in the waterborne coatings (as shown in the red box in Figure 2E,I). The EDX analysis showed that there was no Si in the coatings without WSPALR or calcined WSPALR (Figure 2B), but Si was obvious in the coatings with WSPALR and calcined WSPALR (Figure 2D,F,H,J). As Si is non-volatile and its mass is constant, the Si concentration of waterborne coatings after drying was calculated according to Equation (1):

$$Si \ (\%) = (m_2 Si_{calcined}/m_1) \times 100\% \tag{1}$$

$m_2$ is the quality of calcined coatings, $Si_{calcined}$ is the content of Si measured by EDX analysis of coatings after high-temperature calcination, and $m_1$ is the quality of the coatings before calcination. According to Equation (1), the calculation of Si concentration of waterborne coatings after drying is shown in Table 2. With the increase of WSPALR concentration or high-temperature calcined WSPALR, Si concentration of the waterborne coatings increased, which indicated that the coating can be done by intended content of the fillers. The cross-section SEM images of the waterborne coatings with 5.0% WSPALR and 5.0% calcined WSPALR are shown in Figure 3. The EDX analysis of WSPALR or calcined WSPALR particles in the coating (Figures 2 and 3) has been done and they were similar to those of Figure 1D,F, which indicate the cross-section existence of WSPALR or calcined WSPALR.

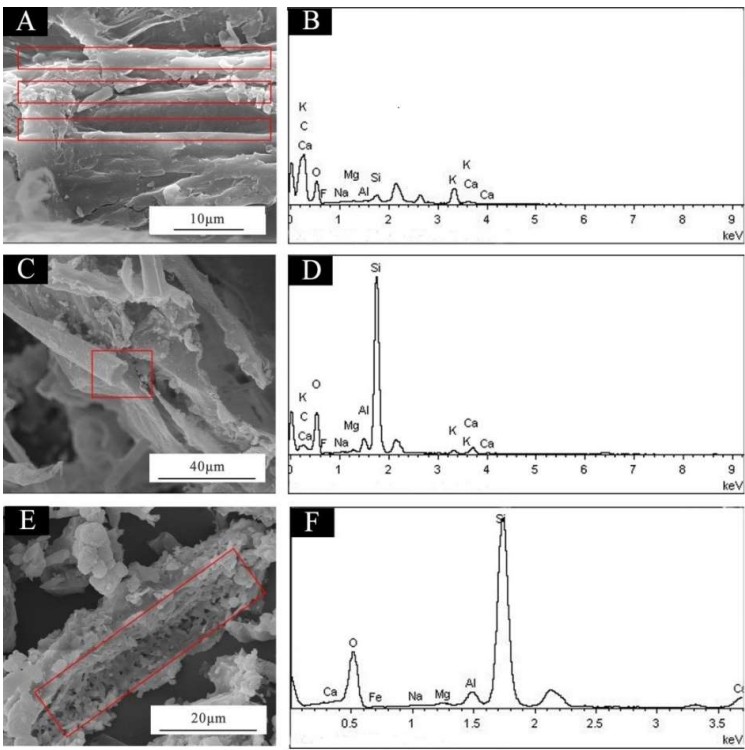

**Figure 1.** SEM images of (**A**) wheat straw powder, (**C**) WSPALR, and (**E**) high-temperature calcined WSPALR. EDX analysis of (**B**) wheat straw powder, (**D**) WSPALR, and (**F**) high-temperature calcined WSPALR.

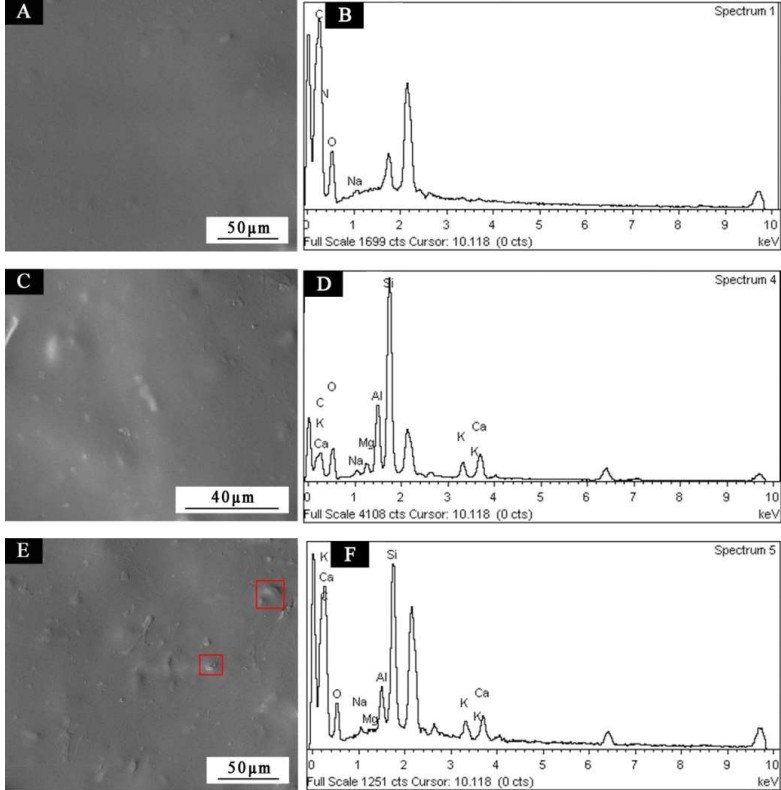

**Figure 2.** *Cont.*

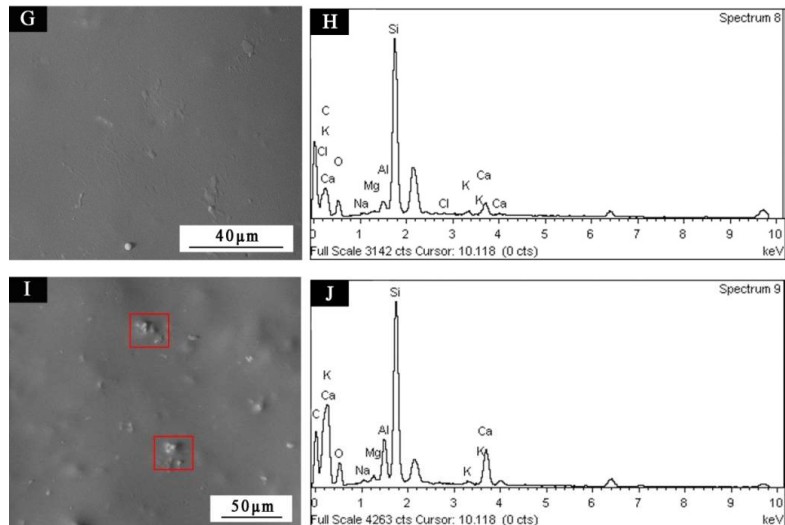

**Figure 2.** SEM images: (**A**) waterborne coating; waterborne coating with different concentrations of WSPALR: (**C**) 0.5% and (**E**) 5.0%; and waterborne coating with different concentrations of calcined WSPALR: (**G**) 0.5% and (**I**) 5.0%. EDX analysis: (**B**) waterborne coating; waterborne coating with different concentrations of WSPALR: (**D**) 0.5% and (**F**) 5.0%; and waterborne coating with different concentrations of calcined WSPALR: (**H**) 0.5% and (**J**) 5.0%.

**Table 2.** Si concentration of waterborne coatings.

| Sample | $m_2$ (g) | $m_1$ (g) | $Si_{calcined}$ (%) | $Si_{coating}$ (%) |
|--------|-----------|-----------|---------------------|--------------------|
| 1 | 0 | 0.79 ± 0 | 0 | 0 |
| 2 | 0.01 ± 0 | 0.80 ± 0 | 38.00 ± 0 | 0.46 ± 0.02 |
| 3 | 0.02 ± 0 | 0.82 ± 0 | 38.00 ± 0.01 | 0.91 ± 0.02 |
| 4 | 0.04 ± 0 | 0.84 ± 0.01 | 38.00 ± 0.01 | 1.75 ± 0.04 |
| 5 | 0.07 ± 0 | 0.87 ± 0.01 | 38.00 ± 0.01 | 2.96 ± 0.27 |
| 6 | 0.10 ± 0 | 0.91 ± 0.01 | 38.00 ± 0.01 | 4.06 ± 0.21 |
| 7 | 0.02 ± 0 | 0.81 ± 0 | 38.00 ± 0.02 | 0.71 ± 0.02 |
| 8 | 0.03 ± 0 | 0.82 ± 0 | 38.00 ± 0.01 | 1.39 ± 0.04 |
| 9 | 0.06 ± 0 | 0.84 ± 0 | 38.00 ± 0 | 2.71 ± 0.09 |
| 10 | 0.11 ± 0 | 0.87 ± 0 | 38.00 ± 0.01 | 4.57 ± 0.24 |
| 11 | 0.15 ± 0 | 0.91 ± 0.03 | 38.00 ± 0.01 | 6.29 ± 0.21 |

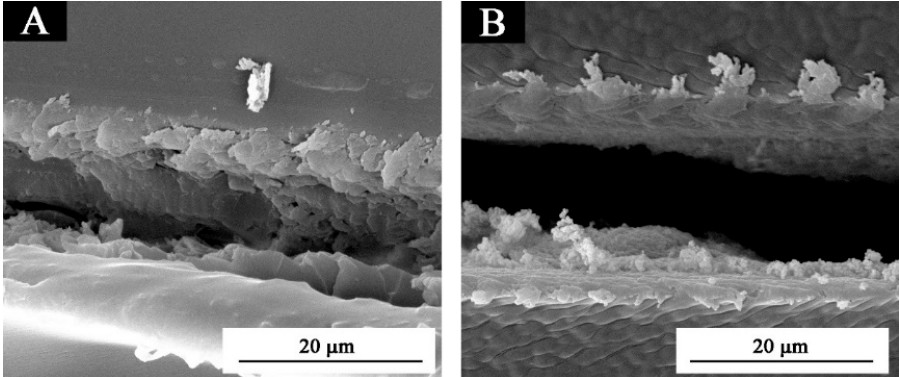

**Figure 3.** The cross-section SEM images of the waterborne coatings with (**A**) 5.0% WSPALR and (**B**) 5.0% calcined WSPALR.

Figure 4 is the infrared spectra of wheat straw powder untreated and treated with alkaline hydrogen peroxide. The absorption band at 3350 cm$^{-1}$ is assigned to –OH stretching vibration, mainly from cellulose, hemicellulose, polysaccharides, and monosaccharides [32]. Bands at 2935 and 2850 cm$^{-1}$ are the stretching vibration bands of –CH$_3$ and –CH$_2$ groups in cellulose [33]. That at 1735 cm$^{-1}$ is the C–O stretching vibration band of carboxylic acids and ketones related to lignin or hemicelluloses [34]. The 1160 cm$^{-1}$ band is the C–O stretching vibration, and the C–C skeleton stretching vibration absorption band of cellulose intramolecular ether, which is the characteristic band of cellulose structure [35]. The band at 1060 cm$^{-1}$ is the Si–O stretching vibration band of inorganic materials such as SiO$_2$ [36]. Compared with untreated wheat straw powder, the C–O stretching vibration band and the C–C skeleton stretching vibration absorption at 1160 cm$^{-1}$ increased after the alkaline hydrogen peroxide, which indicated that the purity of SiO$_2$ and cellulose was improved because the removal of lignin leaves a large amount of cellulose and SiO$_2$ after the reaction. The infrared spectrum of the waterborne coatings added with WSPALR and calcined WSPALR were basically unchanged (Figure 5), indicating that WSPALR or calcined WSPALR and waterborne coatings were physically bonded.

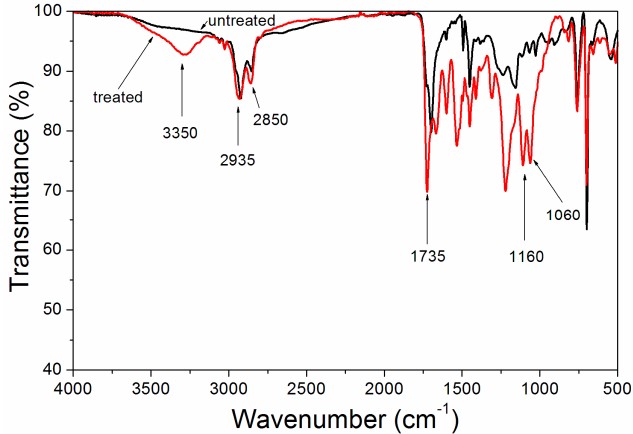

**Figure 4.** The infrared spectrum of wheat straw powder: untreated and treated with alkaline hydrogen peroxide.

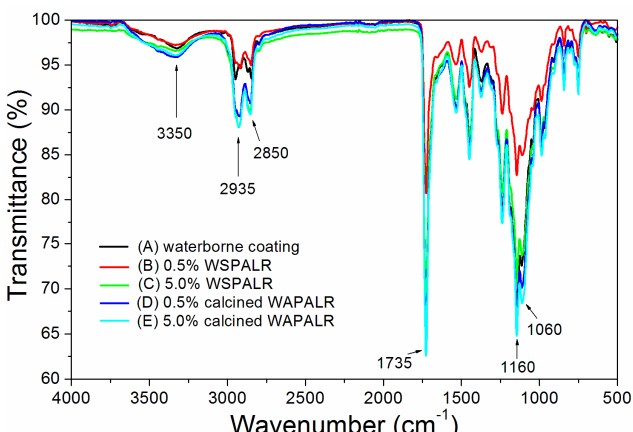

**Figure 5.** The infrared spectrum: (**A**) waterborne coating; waterborne coating with different concentrations of WSPALR: (**B**) 0.5% and (**C**) 5.0%; and waterborne coating with different concentrations of calcined WSPALR: (**D**) 0.5% and (**E**) 5.0%.

Surface hardness is one of many performance criteria of coatings but depends on the application. Harder surfaces are not always better than softer ones. The hardness of the coating is the ability of the coating to resist the external mechanical effects such as collision, scratching, and so on. It is also an important index of whether the performance of the coating is good or not. The effect of WSPALR and high-temperature calcined WSPALR on the hardness of waterborne coatings is shown in Figure 6. According to the trend shown in Figure 6, with the increase of WSPALR concentration, the hardness of the waterborne coatings first increased and then decreased. When the concentration of WSPALR increased from 0 to 1.0%, the hardness of the coating increased to the highest value, which was 6H. When the concentration of WSPALR was 1.0%–2.0%, the hardness had a high value of 6H. When the concentration of WSPALR was more than 2.0%, the hardness of the coating decreased to 5H. When the concentration of high-temperature calcined WSPALR was increased from 0 to 1.0%, the hardness of the coating increased to the highest value of 4H. When the content of high-temperature calcined WSPALR exceeded 1.0%, the hardness of the coating decreased to 3H. The results showed that the hardness of waterborne coatings was the highest when the concentration of WSPALR was 1.0%–2.0% and the concentration of high-temperature calcined WSPALR was 0.5%–1.0%. A small amount of agglomeration of particles occurred due to the large amount of WSPALR and calcined WSPALR (Figure 2). The distribution was uneven, leading to a decrease of hardness [37]. The hardness of waterborne coatings can be improved by adding WSPALR and high-temperature calcined WSPALR, and the hardness of waterborne coatings with WSPALR was higher than that of high-temperature calcined WSPALR. This is because after calcination of WSPALR at a high temperature, WSPALR decomposed and changed its structural morphology from long rods to a porous structure with micropores, and the multi-void structure was not conducive to the improvement of the hardness of the coating.

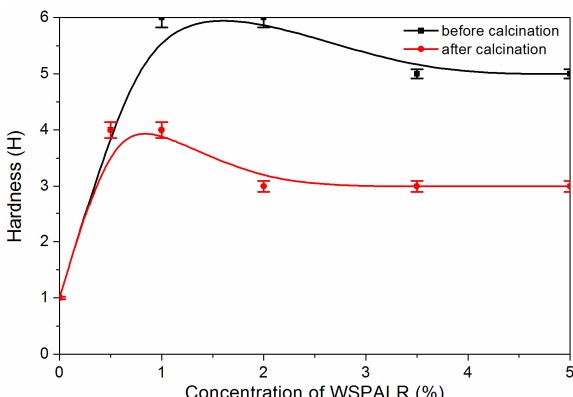

**Figure 6.** Effect on WSPALR and high-temperature calcined WSPALR on the hardness of coatings.

The adhesion of waterborne coatings refers to the ability to form strong bonding between coatings and poplar substrates. Putty was used to fill the vessels of poplar wood and did not increase coating thickness. Coatings with good adhesion can play the role of protection and decoration. Figure 7 shows the effect of WSPALR and high-temperature calcined WSPALR on the adhesion of coatings. It was found that the trend of the adhesion curve of waterborne coatings with the addition of WSPALR and high-temperature calcined WSPALR was basically the same, indicating that the effect of WSPALR and high-temperature calcined WSPALR on the adhesion of waterborne coatings was similar. As can be seen from Figure 7, when the concentration of WSPALR increased from 0 to 2.0%, the adhesion of the coating decreased to Level 4. When the concentration of WSPALR increased from 2.0% to 5.0%, the adhesion of the coating increased from Level 4 to Level 2. From Figure 7, when the concentration of high-temperature calcined WSPALR increased from 0 to 1.0%, the adhesion of the coating decreased to Level 4. When the concentration of high-temperature calcined WSPALR increased from 1.0% to 3.5%, the adhesion of the coating increased from Level 4 to Level 2. When the concentration of WSPALR after high-temperature calcination increased from 3.5% to 5.0%, the adhesion of the coating

was unchanged. From the results shown in Figure 7, when WSPALR and high-temperature calcined WSPALR were added at a concentration of 0.5%, the adhesion of waterborne coating was better, which was Level 1. After sodium hydroxide treatment, lignin was removed on the surface of the wheat straw powder, and also certain hemicellulose and other substances were removed, which was not conducive to enhancing interface bonding with waterborne coatings, and when the concentration of WSPALR and high-temperature calcined WSPALR was higher, it aggregated and did not disperse uniformly, so adding a certain amount of WSPALR and high-temperature calcined WSPALR decreased the adhesion of waterborne coatings.

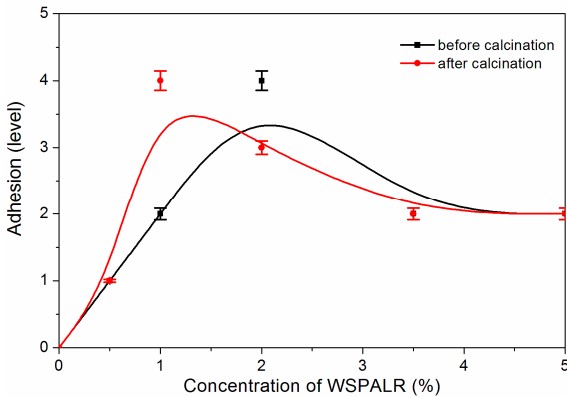

**Figure 7.** Effect of WSPALR and high-temperature calcined WSPALR on adhesion of coatings.

Resistance to impact refers to the ability of the coatings on the substrates to deform under high-speed gravity without cracking or falling off from the substrates. Figure 8 shows the effect of WSPALR and after high-temperature calcination and the resistance to the impact of the coatings. As can be seen from Figure 8, when the concentration of WSPALR increased from 0 to 2.0%, the resistance to impact increased to 12.0 kg cm. When the concentration of WSPALR increased from 2.0% to 5.0%, the resistance to impact decreased from 12.0 to 10.0 kg cm. Figure 8 shows that when the concentration of WSPALR after high-temperature calcination increased from 0 to 1.0%, the resistance to impact increased to 12.0 kg cm. When the concentration of WSPALR after high-temperature calcination increased from 1.0% to 2.0%, the resistance to impact decreased from 12.0 to 11.0 kg cm. When the concentration of WSPALR after high-temperature calcination increased from 2.0% to 5.0%, the resistance to impact decreased from 11.0 to 10.0 kg cm. The data in Figure 8 indicate that its overall fluctuation range was very small. The particles were prone to partial agglomeration at the high concentration of WSPALR and calcined WSPALR (Figure 2), which will cause uneven distribution and decrease the resistance to impact when the concentration of WSPALR was above 2.0% (Figure 8) [37]. The resistance to impact of the waterborne coatings was basically unchanged with an increase in WSPALR concentration from 0.5% to 5.0%, which indicated that the resistance to impact of the waterborne wood coatings could be increased by adding WSPALR and after high-temperature calcination because they can absorb more impact energy [38].

The elongation at break of the coatings is shown in Figure 9. With the increase of the concentration of WSPALR and high-temperature calcined WSPALR, the elongation at break of the coating first increased and then decreased. When the concentration of WSPALR increased from 0 to 1.0%, the elongation at break increased from 2.6% to 5.1% for reinforcement with cellulose fibers leading to significantly higher values of fracture strain [39]. When the concentration of WSPALR increased from 1.0% to 5.0%, the elongation at break decreased from 5.1% to 2.5%. When the concentration of high-temperature calcined WSPALR increased from 0 to 1.0%, the elongation at break increased from 2.6% to 4.8% for the improved toughening induced by the ash particles (ash comes from the calcined WSPALR) [40]. When the concentration of high-temperature calcined WSPALR increased from 1.0% to 5.0%, the elongation at break decreased from 4.8% to 2.4%. The results showed that the elongation at break of the coating

was the highest when the concentration of WSPALR and high-temperature calcined WSPALR was 1.0%. The particles were prone to partial agglomeration at the high concentration of WSPALR and calcined WSPALR (Figure 2), therefore, the elongation at break of the coating decreased.

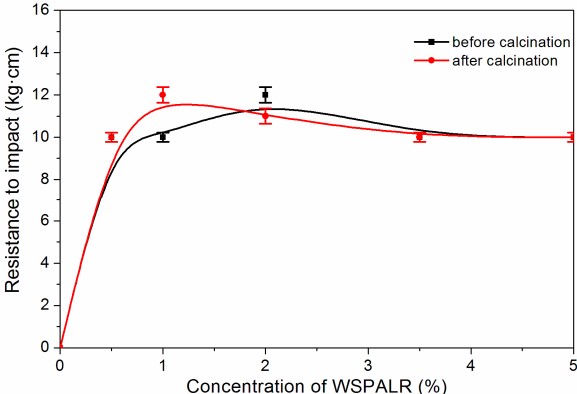

**Figure 8.** Effect of WSPALR and high-temperature calcined WSPALR on the resistance to impact of coatings.

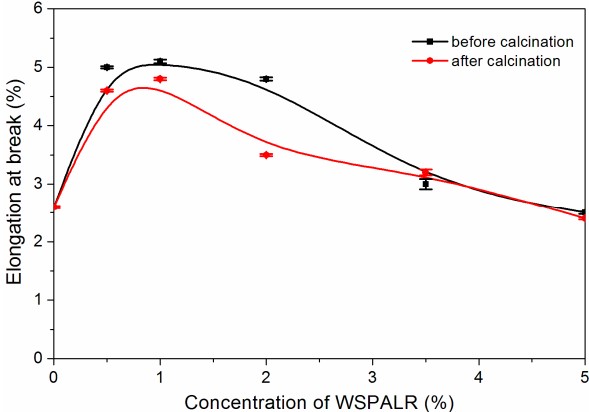

**Figure 9.** Effect of WSPALR and high-temperature calcined WSPALR on the elongation at break of coatings.

The color difference of coatings refers to the difference in color when the light source is polychromatic light. $L$, $a^*$, $b^*$ respectively represent the black-white, red-green, yellow-blue values at any position of the waterborne wood coatings. $L'$, $a^{*'}$, $b^{*'}$ respectively represent the black-white, red-green, yellow-blue values at another location of the waterborne wood coatings. After subtraction, the difference values $\Delta L$, $\Delta a^*$, $\Delta b^*$, respectively, are expressed as the lightness difference, red-green color difference, and yellow-blue color difference. Thus, the color difference $\Delta E$ can be obtained according to Equation (2):

$$\Delta E = \sqrt{(\Delta L)^2 + (\Delta a^*)^2 + (\Delta b^*)^2} \tag{2}$$

Tables 3 and 4 show the effects of WSPALR and high-temperature calcined WSPALR on the color difference of waterborne coatings. The effect of color difference on the acceptability of waterborne coatings is shown in Table 5 [41]. From the data in Table 3, when WSPALR was not added, the color difference of the coating was very small and uniform. When WSPALR was added, the color difference of the coating increased by varying degrees. WSPALR and calcined WSPALR have color (Figure 10), which may also affect the color difference. When the concentration of WSPALR was 5.0%, the color difference of the coating increased significantly, reaching the maximum value of 8.4. This is because when the concentration of WSPALR was higher, it was not easy to disperse (Figure 2) due

to the compatibility between the coating and WSPALR. Similarly, from Table 4, with the increase of WSPALR concentration after calcination at high temperature, the color difference of waterborne coatings increased. Although the color difference of the coatings increased, the increase was not obvious, indicating that the addition of high-temperature calcined WSPALR could maintain the color difference of the original coatings. This is because when the concentration of high-temperature calcined WSPALR was higher, it aggregated and did not disperse uniformly (Figure 2). The results showed that the addition of high-temperature calcined WSPALR had little effect on the color of waterborne coatings, and the color of waterborne coatings was more uniform. This is due to the carbonization of WSPALR after calcination at high temperature and the formation of granular products, which are finer than those before calcination. The addition of a low concentration of WSPALR and high-temperature calcined WSPALR was applicable and acceptable for waterborne coatings despite its small color difference.

The results of mold resistance of the coating are shown in Figure 11. In Figure 11, it can be seen that Trichoderma grew on the poplar veneers with the WSPALR concentration of 0.5%. Trichoderma did not appear on the poplar veneers coated with the calcined WSPALR. It can be seen that Trichoderma grew on the poplar veneers coated with waterborne coatings without added WSPALR. The Trichoderma generated portion was between the coating and the wood (as shown in the red box in Figure 11A,B). This is because the effect of waterborne coatings on wood mold resistance was not good enough. Compared with other added concentrations, the addition of the lower concentration of WSPALR made it easier for Trichoderma to grow, and the higher concentration of WSPALR had a mold-resistance effect. The more WSPALR that was added, the better the inhibition effect on Trichoderma. This is because WSPALR itself is long and rod-like and accompanied by some tubular substances (Figure 1C), which had an inactivating effect on Trichoderma and afforded a certain degree of mold resistance. When WSPALR was calcined at a high temperature, carbon was basically transformed into carbon dioxide and escaped to form porous and irregular composites with micropores (Figure 1E), which had activity and can prevent the propagation of fungi [26], so that the inhibition effect of high-temperature calcined WSPALR on Trichoderma was better than that of WSPALR.

**Table 3.** Effect of WSPALR on the color difference of waterborne coatings.

| Concentration of WSPALR (%) | $L$ | $a^*$ | $b^*$ | $L'$ | $a^{*'}$ | $b^{*'}$ | $\Delta L$ | $\Delta a^*$ | $\Delta b^*$ | $\Delta E$ |
|---|---|---|---|---|---|---|---|---|---|---|
| 0 | 59.80 ± 0 | 13.00 ± 0.02 | 21.20 ± 0.02 | 59.30 ± 0.01 | 11.80 ± 0.01 | 21.20 ± 0.22 | 0.50 ± 0.01 | 1.20 ± 0.02 | 0 ± 0.21 | 1.30 ± 0.05 |
| 0.5 | 51.40 ± 0 | 13.20 ± 0.08 | 29.80 ± 0.02 | 51.20 ± 0.02 | 12.60 ± 0.03 | 30.60 ± 0 | 0.20 ± 0.02 | 0.60 ± 0.10 | −0.80 ± 0.02 | 1.00 ± 0.05 |
| 1.0 | 52.30 ± 0.04 | 13.20 ± 0 | 24.50 ± 0 | 51.30 ± 0 | 14.30 ± 0.06 | 23.10 ± 0 | 1.00 ± 0.04 | −1.10 ± 0.06 | 1.40 ± 0 | 2.0 ± 0.05 |
| 2.0 | 52.30 ± 0.02 | 14.60 ± 0 | 25.00 ± 0.02 | 52.30 ± 0 | 15.90 ± 0.03 | 25.20 ± 0 | 0 ± 0.02 | −1.30 ± 0.03 | −0.20 ± 0.02 | 1.3 ± 0.05 |
| 3.5 | 50.20 ± 0 | 13.10 ± 0.02 | 29.40 ± 0.02 | 49.80 ± 0.02 | 11.80 ± 0.03 | 29.60 ± 0 | 0.40 ± 0.02 | 1.30 ± 0.05 | −0.20 ± 0.02 | 1.4 ± 0.05 |
| 5.0 | 47.10 ± 0 | 20.00 ± 0.02 | 35.50 ± 0.05 | 47.70 ± 0.02 | 23.70 ± 0.03 | 43.00 ± 0.08 | −0.60 ± 0.02 | −3.70 ± 0.05 | −7.50 ± 0.04 | 8.4 ± 0.05 |

**Table 4.** Effect of high-temperature calcined WSPALR on the color difference of waterborne coatings.

| Concentration of Calcined WSPALR (%) | $L$ | $a^*$ | $b^*$ | $L'$ | $a^{*'}$ | $b^{*'}$ | $\Delta L$ | $\Delta a^*$ | $\Delta b^*$ | $\Delta E$ |
|---|---|---|---|---|---|---|---|---|---|---|
| 0 | 59.80 ± 0 | 13.00 ± 0.02 | 21.20 ± 0.02 | 59.30 ± 0.01 | 11.80 ± 0.01 | 21.20 ± 0.22 | 0.50 ± 0.01 | 1.20 ± 0.02 | 0 ± 0.21 | 1.30 ± 0.05 |
| 0.5 | 63.90 ± 0.02 | 12.70 ± 0.02 | 36.80 ± 0 | 64.70 ± 0 | 12.10 ± 0 | 36.40 ± 0.11 | −0.80 ± 0.02 | 0.60 ± 0.02 | 0.40 ± 0.11 | 1.10 ± 0.05 |
| 1.0 | 66.20 ± 0.02 | 12.40 ± 0 | 30.80 ± 0 | 68.00 ± 0.02 | 10.70 ± 0.02 | 29.10 ± 0 | −1.80 ± 0.04 | 1.70 ± 0.02 | 1.70 ± 0 | 3.00 ± 0.05 |
| 2.0 | 60.70 ± 0.02 | 14.60 ± 0 | 34.00 ± 0.02 | 57.50 ± 0.04 | 14.80 ± 0.02 | 34.50 ± 0 | 3.20 ± 0.02 | −0.20 ± 0.02 | −0.50 ± 0.02 | 3.20 ± 0.05 |
| 3.5 | 69.50 ± 0 | 11.40 ± 0.02 | 33.70 ± 0.02 | 64.80 ± 0.02 | 8.60 ± 0.02 | 32.60 ± 0 | 4.70 ± 0.02 | 2.80 ± 0.04 | 1.10 ± 0.02 | 5.60 ± 0.05 |
| 5.0 | 65.90 ± 0.02 | 12.90 ± 0.02 | 33.50 ± 0 | 60.30 ± 0 | 13.30 ± 0.02 | 34.40 ± 0.11 | 5.60 ± 0.02 | −0.40 ± 0.01 | −0.90 ± 0.11 | 5.70 ± 0.05 |

**Table 5.** Effect of color difference on the acceptability of waterborne coatings.

| $\Delta E$ | Acceptability |
|---|---|
| 0–0.25 | Very small or not: ideal matching |
| 0.25–0.5 | Small: acceptable |
| 0.5–1.0 | Small to the medium: acceptable in some applications |
| 1.0–2.0 | Medium: acceptable in specific applications |
| 2.0–4.0 | Gaps: acceptable in specific applications |
| >4.0 | Very large: basically unacceptable |

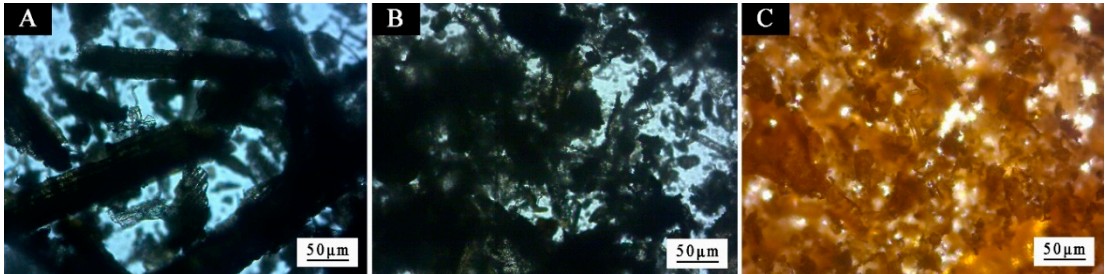

**Figure 10.** Digital microscopic images of (**A**) wheat straw powder, (**B**) WSPALR, and (**C**) high-temperature calcined WSPALR.

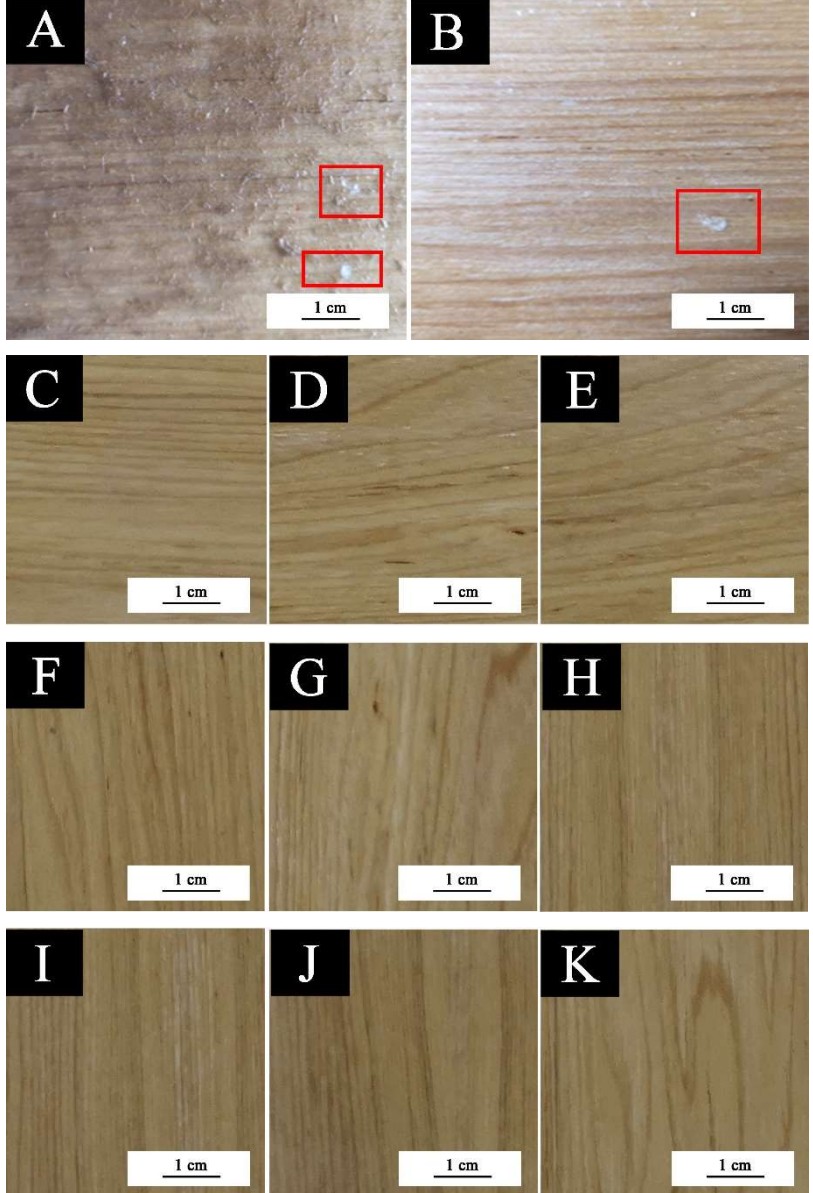

**Figure 11.** Trichoderma grew on the surface of wood substrates: (**A**) waterborne coating; waterborne coating with different concentrations of WSPALR: (**B**) 0.5%; (**C**) 1.0%; (**D**) 2.0%; (**E**) 3.5%; (**F**) 5.0%; and waterborne coating with different concentrations of calcined WSPALR: (**G**) 0.5%; (**H**) 1.0%; (**I**) 2.0%; (**J**) 3.5%; (**K**) 5.0%.

## 4. Conclusions

The experimental results show that the hardness of waterborne coatings with WSPALR and high-temperature calcined WSPALR can be improved. The hardness of waterborne coatings with WSPALR was higher than that of waterborne coatings with WSPALR after calcination at high temperature. The hardness and resistance to the impact of waterborne coatings first increased and then decreased with the increase of the WSPALR concentration. The adhesion of the waterborne wood coatings could be maintained by adding WSPALR and high-temperature calcined WSPALR. The effect of WSPALR calcined at a high temperature on the color of waterborne coatings was smaller, and the color of waterborne coatings was more uniform. The higher concentration of WSPALR had better mold resistance. High-temperature calcined WSPALR had a good inhibitory effect on Trichoderma and was better than that of WSPALR. In summary, when the concentration of WSPALR calcined at a high temperature was 0.5%, the hardness of the waterborne coatings was 4H, adhesion of the waterborne coatings was Level 1, the resistance to the impact of the waterborne coatings was 10 kg cm, the elongation at break was 4.6%, the color difference of the waterborne coatings was 1.1, and it had better mold resistance. At this time, the comprehensive performance of the coating was better. The concentration of 0.5% for WSPALR calcined at a high-temperature in the coating was a better concentration, which can stand for another preparation of coating methods. This paper provides a good basis for the application of waterborne wood coatings in wood surface resistance to Trichoderma.

**Author Contributions:** Conceptualization, Methodology, Validation, Resources, Data Curation, Writing—Original Draft Preparation, and Supervision, X.Y.; Formal Analysis, L.W. and X.Q.; Investigation and Writing—Review and Editing, L.W.

**Funding:** This research was funded by the Natural Science Foundation of Jiangsu Province (No. BK20150887) and Youth Science and Technology Innovation Fund of Nanjing Forestry University (No. CX2016018).

**Conflicts of Interest:** The authors declare no conflict of interest.

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
