# Peer review of "Effect of High-Temperature Calcined Wheat Straw Powder after Lignin Removal on Properties of Waterborne Wood Coatings"

_coatings, doi:10.3390/coatings9070444_

Reviewer 1 Report

The authors must improve the abstract, to include quantitative (not descriptive) results regarding the properties of the coating

The authors still didn't include the reason for using the calcined cellulose, as the first recommendation suggested 

Page 2 line 84- what properties are we talking about?

The viscosity of the waterborne wood coating is 20 s.- this must be wrong. Convert it to viscosity units

What is the role of the putty? 

The authors still did not improve the description of the experimental methodology for coatings obtaining, as it was suggested in the first place.

What type of hardness does graph 5 illustrate? Measurement units? Error bars?

How was the adhesion (level) measured? What type of physical quantity is on the Y axis? Error bars? 

The "before" and "after" statements are confusing

The paper does not describe the methodology  for mould resistance testing.

Author Response

The reviewers give very valuable comments on our manuscript, and we would like to take this opportunity to express our great appreciation on him/her as well as the comments. Followings are our responses to the reviewers’ questions and comments together with the changes made in the revised version.

Q1:The authors must improve the abstract, to include quantitative (not descriptive) results regarding the properties of the coating.

A:The abstract has been improved and the quantitative results regarding the properties of the coating have been include. It is in the text on page 1, line 15-24 in the revised version, thank you very much.

Q2:The authors still didn’t include the reason for using the calcined cellulose, as the first recommendation suggested.
A:The reason for using the calcined cellulosehas been added. It is in the text on page 2, line 74-77 in the revised version, thank you very much.

Q3: Page 2 line 84- what properties are we talking about?
A:The properties we are talking about are color difference, hardness, adhesion, and resistance to impact. The properties of waterborne coatings were analyzed to improve the mould resistance of waterborne coatings while maintaining their color difference, hardness, adhesion, and resistance to impact properties. It is in the text on page 2, line 84 in the revised version, thank you very much.

Q4: The viscosity of the waterborne wood coating is 20 s.- this must be wrong. Convert it to viscosity units.
A:The viscosity unit has been changed. It is in the text on page 3, line 96 in the revised version, thank you very much.

Q5: What is the role of the putty?
A:The main ingredients of putty are talc powder and water, and the role of putty was to fill the vessels of poplar wood. It is in the text on page 3,line 114 in the revised version, thank you very much.

Q6: The authors still did not improve the description of the experimental methodology for coatings obtaining, as it was suggested in the first place.
A: The description of the experimental methodology for coatings obtaining has been improved. The main ingredients of putty are talc powder and water, and the role of putty was to fill the vessels of poplar wood. 80.0g talc powder was added to 200.0g water and mixed evenly to obtain the putty. Then the putty was coated evenly on the poplar as a veneer. When the putty was dry, the poplar veneer was sanded using 600 grit sandpaper, and a dry cloth was used to wipe off the dust. The waterborne wood coating was sprayed using an airbrush (Guangzhou Zhongtian Electrical Equipment Co. Ltd., Guangzhou, China) on poplar veneers, waiting 30 min for natural drying, and was sanded using 1000 grit sandpaper; then a dry cloth was used to wipe off the dust. The spray process of waterborne wood coating needed to be repeated twice. 0.5g WSPALR was added to 99.5g waterborne wood coatings to form 0.5% concentration and mixed evenly. The waterborne coatings with 1.0%, 2.0%, 3.5%, and 5.0% concentrations of WSPALR were also prepared. The formulations are summarized in Table 1.
This answer is incorporated in the text on page 3,line 115-134 in the revised version, thank you very much.

Q7: What type of hardness does graph 5 illustrate? Measurement units? Error bars?
A: Graph 5 illustrates the hardness of the film determined with 6H, 5H, 4H, 3H, 2H and 1H pencils. Measurement unit is H. Error bars have been shown in Figure 5. It is incorporated in the text on page 4 line 157 and on page 7 line 236 in the revised version, thank you very much.

Q8: How was the adhesion (level) measured? What type of physical quantity is on the Y axis? Error bars?
A:According to GB/T 1720-89 [28], a QFZ-II circle-cut coating adhesive testing machine (Tianjin JingKelian Material Testing Machine Co., Ltd.) was used to determine the adhesion of the coating. The experiment was carried out at 23±2C and 50±5% relative humidity. When measuring, the tip of the rotary needle touches the coating, and the handle is shaken uniformly clockwise. The rotational speed is in the range of 80-100 r/min, and the scratch length of the circular roller is 7.5±0.5 mm. At the end of the adhesion test, the damage to the coating was observed using a magnifier. The marks 0, 1, 2, 3, 4, 5, 6, and 7 on the top of the coating are the 8 levels of damage. The adhesion of level 0 was the strongest, and that of level 7 was the weakest.The physical quantity on the Y axis is level. Error bars have been shown in Figure 6.
This answer is incorporated in the text on page 4 line 150-152 and Figure 6 in the revised version, thank you very much.

Q9: The "before" and "after" statements are confusing.
A:The "before" has been deleted.
This answer is incorporated in the text on page 7 line 217, 231, 232, 241, 243, and on page 8 line 253 and 259 in the revised version, thank you very much.

Q10:The paper does not describe the methodology for mould resistance testing.
A:The methodology for mould resistance testing has been added. The samples were placed in an incubator. The bottom of the incubator has enough exposed water surface area to ensure that the sample is not touched or splashed by water and that the samples are not placed in contact with each other. The temperature was between 25°C and 30°C and relative humidity was 85.0%. After 5 d, mould growth on the surface was observed. This answer is incorporated in the text on page 4 line 160-162 in the revised version, thank you very much.

PS: All the revised parts in the manuscript were highlighted by green color.

Reviewer 2 Report

The authors have improved the quality of their paper.

However, although the authors claim to have used 300 samples with four replicate mesaurements, only few data points are presented in their study. This makes it very hard to see the significance of their results.

Furthermore, the SEM images in Figure 2 show a very uneven distribution of the WSPALR particles. This makes it hard to believe that this will have an positive effect on the impact resistance of the coatings.

Author Response

The reviewers give very valuable comments on our manuscript, and we would like to take this opportunity to express our great appreciation on him/her as well as the comments. Followings are our responses to the reviewers’ questions and comments together with the changes made in the revised version.

Q1:However, although the authors claim to have used 300 samples with four replicate measurements, only few data points are presented in their study. This makes it very hard to see the significance of their results.
A:We have measured more than 300 samples and summarized the results in this paper. In order to give readers a clearer understanding of these results, and Error bars have also been added to show the significance of the results. It is in the text on page 7 line 236, on page 8 line 260, on page 9 line 280, on page 10 line 290, and on page 10 line 292 in the revised version, thank you very much.

Q2:Furthermore, the SEM images in Figure 2 show a very uneven distribution of the WSPALR particles. This makes it hard to believe that this will have a positive effect on the impact resistance of the coatings.
A:The impact resistance of the coatings has been revised. It is in the text on page 8 line 266-267 and on page 8 line 274-275 in the revised version, thank you very much.

PS: All the revised parts in the manuscript were highlighted by green color.

Reviewer 3 Report

This article examined usefulness of wheat straw as a bio-filler for a waterborne coating. The aim and concept are well understood. However there might be some unclear and doubtful contents in the experimental procedure and results.

Most suspicious point is whether the coating can be done by intended content quantities of the fillers. In this study, coating of mixture of patty and water is done by spray using an airbrush. Depending on the size, shape and compatibility between fillers and painting liquids, spraying could have trouble. As can be seen in SEM views of the coating surface, coating with and without filler do not give big difference even in the content changes. Does this indicate the coating cannot be obtained as you intend ?

The existence of fillers in the resultant coating should be confirmed by magnified SEM images as well as FTIR measurements. EDX Si-imaging might be give us useful information. Further in the FTIR quantitative analysis, inner standard must be needed otherwise it is meaningless to discuss Figures 5 - 8 and Tables 2-4.

In the experimental procedure of the SEM and FTIR, how did you observe or measure ?

Those were done on the coating on wood or coating only, or use of an ATR method? Please give us more information.

Also about the Chinese standards showing reference 27-30, it is better to add schematic drawings of photos in the supplemental information.

Author Response

The reviewer gives very valuable comments on our manuscript, and we would like to take this opportunity to express our great appreciation on him/her as well as the comments. Followings are our responses to the reviewer’s questions and comments together with the changes made in the revised version.

Q1: Most suspicious point is whether the coating can be done by intended content quantities of the fillers. In this study, coating of mixture of patty and water is done by spray using an airbrush. Depending on the size, shape and compatibility between fillers and painting liquids, spraying could have trouble. As can be seen in SEM views of the coating surface, coating with and without filler do not give big difference even in the content changes. Does this indicate the coating cannot be obtained as you intend ?

A: During the preparation of the coating, in addition to spraying with an airbrush, the coatings were also prepared by using SZQ tetrahedral fabricator. The coating can be done by intended content quantities of the fillers. The waterborne wood coating was coated using SZQ tetrahedral fabricator (Tianjin Jinghai Science and Technology Testing Machinery Factory, Tianjin, China) on poplar veneers. The veneer was fixed on the platform. Then the prepared coatings were poured in front of the fabricator. The two ends of fabricator were grasped by hand, then glided at a uniform speed of 150 mm/s, and the required coating can be coated.

This answer is incorporated in the text on page 3, line 118-122 in the revised version, thank you very much.

Q2: The existence of fillers in the resultant coating should be confirmed by magnified SEM images as well as FTIR measurements. EDX Si-imaging might be give us useful information. Further in the FTIR quantitative analysis, inner standard must be needed otherwise it is meaningless to discuss Figures 5 - 8 and Tables 2-4.

A: The existence of fillers in the resultant coating has been confirmed by SEM and EDX analysis. The SEM image and EDX analysis of the waterborne coatings with WSPALR and calcined WSPALR added are shown in Figure 2, indicating that a small amount of agglomeration of WSPALR or calcined WSPALR occurs due to the large numbers of particles in the waterborne coatings (Figure 2(E) and (I)). The EDX analysis showed that there was no Si in the coatings without WSPALR or calcined WSPALR (Figure 2(B)), but Si is obvious in the coatings with WSPALR and calcined WSPALR (Figure 2(D), (F), (H) and (J)). Because we did not add inner standard materials in FTIR, we could not make quantitative analysis by inner standard method. However, the concentration of Si in the coating was calculated by EDX analysis, which proved the existence of fillers in the coating. The coatings were placed in a high-temperature calciner at a heating rate of 5°C/min, heated to 920°C and kept for 45 min to obtain high-temperature calcined coating for EDX analysis to calculate the Si concentration. Because Si is non-volatile and its mass is constant, the Si concentration of waterborne coatings after drying was calculated according to the formula (1):

Si (%)=(m2·Sicalcined/m1)·100%                    (1)

m2 is the quality of calcined coatings, Sicalcined is the content of Si measured by EDX analysis of coatings after high temperature calcination, and m1 is the quality of coatings before calcination. According to formula (1), the calculation of Si concentration of waterborne coatings after drying is shown in Table 2. With the increase of WSPALR concentration or high-temperature calcined WSPALR, Si concentration of the waterborne coatings increased, which indicated that the coating can be done by intended content quantities of the fillers.

This answer is incorporated in the text on page 4, line 145-148, page 5, line 192-205, and page 6, Figure 2, Table 2 in the revised version, thank you very much.

Q3: In the experimental procedure of the SEM and FTIR, how did you observe or measure?

A: For SEM observation, the samples are adhered to the sample pedestal, and the appropriate time and voltage of gold plating are set for gold plating. After the gold plating is finished, the samples are taken out and placed in the sample chamber. When the vacuum degree reaches a certain value, high pressure is added and the SEM of samples was observed.

For FTIR measurement, wheat straw powder and WSPALR are observed by pressing method. 1~2 mg powder and 200 mg pure KBr are mixed evenly and placed in the mould. The sample is pressed into transparent tablets for testing. The coatings on wood surface are made by ATR method. The samples are placed above the ATR accessory to make the samples contact closely with the ATR crystal. The infrared beam attenuates and reflects in the ATR crystal (diamond) and reaches the detector.

This answer is incorporated in the text on page 4, line 142-145 and line 149-154 in the revised version, thank you very much.

Q4: Those were done on the coating on wood or coating only, or use of an ATR method? Please give us more information.

A: The coatings on wood surface are made by ATR method. The samples are placed above the ATR accessory to make the samples contact closely with the ATR crystal. The infrared beam attenuates and reflects in the ATR crystal (diamond) and reaches the detector.

This answer is incorporated in the text on page 4, line 152-154 in the revised version, thank you very much.

Q5: Also about the Chinese standards showing reference 27-30, it is better to add schematic drawings of photos in the supplemental information.

A: The schematic drawings of photos about the Chinese standards showing reference 27-30 have been added in the supplemental information.

Reference 27: GB/T 1723-1993 Viscosimetry of Coatings; Standardization Administration of the People’s Republic of China: Beijing, China, 1993; pp. 382–385. (In Chinese).

Ford cup No. 4

Reference 28: GB/T 1732-93 Determination of Impact Resistance of Film; Standardization Administration of the People’s Republic of China: Beijing, China, 1993; pp. 418–420. (In Chinese).

QCJ impactor

Reference 29: GB/T 1720-89 Determination of Adhesion of Film; Standardization Administration of the People’s Republic of China: Beijing, China, 1979; pp. 378–379. (In Chinese).

Reference 30: GB/T 1741-2007 Test method for determining the resistance of paints film to mold; Standardization Administration of the People’s Republic of China: Beijing, China, 2007; pp. 1–6. (In Chinese).

PS: All the revised parts in the manuscript were highlighted by blue color.

Round  2

Reviewer 1 Report

The reviewers have improved the paper and I think it could be accepted for publication in Coatings.

Reviewer 2 Report

The authors have improved their paper according to the reviewers suggestions.

Reviewer 3 Report

Additional information was seen. Still there are many points that I do not know yet.

First, some characterization data should be explained in addition to SEM observations of cooked wheat straws in comparison of the control. It must be mentioned how much degree you cooked wheat straws, indicating mass losses derived from delignification, calcination. Because you are discussing porosity or voids of cooked wheat straws, it is better to add information about bulk density at least (or relative density).

○Why did you chose the temperature of 920 deg. C for your calcination ? Please add references.

Present SEM images could not show any void structure as well as whole aggregating fiber cell structure of wheat straws.

○Could you change the SEM images showing hierarchy structures instead of present ones ?

In your study, Si concentration (like in table 2), dispersibility of cooked wheat straws and affinity between waterborne coating and cooked wheat straws may be important. From these points of view, additional magnified SEM images of coating surfaces should be added so as to distinguish boundary of paint and wheat straws. Further it better to show cross sectional SEM views of the coated wood surfaces to know cross sectional existence of wheat straws. Since you are also discussing color changes, digital microscopic image of wheat straws showing their colors better to be added.

○Could you re-consider present x-axes of Fig. 5-7 for more suit parameters like Si-concentration, porosity of coating and density in a further discussion ?

○Did you do EDX imaging measurements for wheat straws dispersed in the coating ?  

○Did you measure some mechanical properties of neat paint film (coating layer only) with and without filler, not for coated wood surface ? Is it difficult to be performed ?

○Were the coated wood samples swelling in moist condition like mould resistance test ?

If so, coating might break due to reaching an ultimate tensile strength.

○How much can the waterborne coating prevent wood from swelling ?

○Where the Trichoderma came from ?  inside poplar wood ? 

In Figure 8 I could not recognize the Trichoderma generate portion. Did It grow between coating and wood ? Please point out the place.

To conclude an optimum condition for something, there must be extremely narrow state of interest. Do you think your optimum concentration of 0.5% for WSPALR calcined at high temperature in coating can stand for another preparation of coating methods ?

It must be careful for use the term “optimum”.

An English proof reading by a native speaker should be recommended because there are some misspellings, strange and confused sentences and words.

Author Response

Reviewer 3

The reviewer gives very valuable comments on our manuscript, and we would like to take this opportunity to express our great appreciation on him/her as well as the comments. Followings are our responses to the reviewer’s questions and comments together with the changes made in the revised version.

Q1: First, some characterization data should be explained in addition to SEM observations of cooked wheat straws in comparison of the control. It must be mentioned how much degree you cooked wheat straws, indicating mass losses derived from delignification, calcination. Because you are discussing porosity or voids of cooked wheat straws, it is better to add information about bulk density at least (or relative density).

A: The weight loss of wheat straw powder after hydrogen peroxide and sodium hydroxide treatment was 47.0%, and that of WSPALR after high temperature calcination was 35.4%. The tap bulk density of wheat straw powder, WSPALR and WSPALR after high temperature calcination was 200.0±5.0 kg/m3, 300.0±5.0 kg/m3 and 800.0±5.0 kg/m3, respectively.

This answer is incorporated in the text on page 3, line 115-118 in the revised version, thank you very much.

Q2: Why did you choose the temperature of 920 deg. C for your calcination? Please add references.

A: Calcination at 920°C was selected for the full volatilization of carbon and hydrogen in WSPALR, which can help improve the properties of coatings according to Ref. 28.

This answer is incorporated in the text on page 3, line 111-112 in the revised version, thank you very much.

Q3: Present SEM images could not show any void structure as well as whole aggregating fiber cell structure of wheat straws.

A: Wheat straws is not void structure and the SEM image of wheat straw powder showed a layered structure (as shown in the red box in Figure 1(A)). Calcined WSPALR is porous and showed the multi-void structure (as shown in the red box in Figure 1(E)).

This answer is incorporated in the text on page 5, line 196 and line 207 in the revised version, thank you very much.

Q4: Could you change the SEM images showing hierarchy structures instead of present ones?

A: The magnified SEM image has been added to show hierarchy structures (as shown in the red box in Figure 1(A)).

This answer is incorporated in the text on page 5, line 196 and Figure 1(A) in the revised version, thank you very much.

Q5: In your study, Si concentration (like in table 2), dispersibility of cooked wheat straws and affinity between waterborne coating and cooked wheat straws may be important. From these points of view, additional magnified SEM images of coating surfaces should be added so as to distinguish boundary of paint and wheat straws. Further it better to show cross sectional SEM views of the coated wood surfaces to know cross sectional existence of wheat straws. Since you are also discussing color changes, digital microscopic image of wheat straws showing their colors better to be added.

A: In this study, wheat straws were not added to the coatings, so it is not necessary to distinguish the boundary of paint and wheat straws. Furthermore, the magnified SEM images of coating surfaces (as shown in the red box in Figure 2(E) and (I)) have been added to distinguish boundary of paint and WSPALR or calcined WSPALR. The cross-section SEM images of the waterborne coatings with 5.0% WSPALR and 5.0% calcined WSPALR have been added as shown in Figure 3. The EDX analysis of WSPALR or calcined WSPALR particles in Figure 3 have been done and they are similar to those of Figure 1(D) and (F), which indicate the cross-section existence of WSPALR or calcined WSPALR. The digital microscopic images of wheat straw powder, WSPALR, and high-temperature calcined WSPALR were observed by Zeiss AXIO Scope A1 microscope (Carl Zeiss AG, Oberkochen, Germany). The digital microscopic images of wheat straw powder, WSPALR, and high-temperature calcined WSPALR have been added in Figure 10 and showed different colors.

This answer is incorporated in the text on page 4, line 179-181, page 5, line 210, 221-225, page 13, line 368 and Figure 10 in the revised version, thank you very much.

Q6: Could you re-consider present X-axes of Fig. 5-7 for more suit parameters like Si-concentration, porosity of coating and density in a further discussion?

A: WSPALR or calcined WSPALR is a composite material, not a single material. The properties of the coatings depend on the concentration of the WSPALR or calcined WSPALR, therefore, it’s better to use concentration of material itself as the X-axis and the X-axes could not be replaced by Si-concentration, porosity of coating and density.

Q7: Did you do EDX imaging measurements for wheat straws dispersed in the coating?

A: It is not wheat straws that are added to the coating, but WSPALR or calcined WSPALR. The EDX analysis of WSPALR or calcined WSPALR particles in the coating has been done and they are similar to those of Figure 1(D) and (F).

This answer is incorporated in the text on page 5, line 223 in the revised version, thank you very much.

Q8: Did you measure some mechanical properties of neat paint film (coating layer only) with and without filler, not for coated wood surface? Is it difficult to be performed?

A: The waterborne coatings with and without WSPALR and high-temperature calcined WSPALR were prepared onto the glass substrates according to the same process and the coating thickness was about 40 μm for tensile test measurement. The elongation at break of the coatings has been tested by the Model AG-IC100KN precision electronic universal capability experiment machine (Shimadzu Co., Ltd., Kyoto, Japan). The coating was peeled off the glass substrate and the elongation at break of the coating was calculated according to the displacement length of the coating at break and the original length of the coating before stretching. The elongation at break curve of the coatings was shown in Figure 9. With the increase of the concentration of WSPALR and high-temperature calcined WSPALR, the elongation at break of the coating first increased and then decreased. When the concentration of WSPALR increased from 0 to 1.0%, the elongation at break increased from 2.6% to 5.1%. When the concentration of WSPALR increased from 1.0% to 5.0%, the elongation at break decreased from 5.1% to 2.5%. When the concentration of high-temperature calcined WSPALR increased from 0 to 1.0%, the elongation at break increased from 2.6% to 4.8%. When the concentration of high-temperature calcined WSPALR increased from 1.0% to 5.0%, the elongation at break decreased from 4.8% to 2.4%. The results showed that the elongation at break of the coating was the highest when the concentration of WSPALR and high-temperature calcined WSPALR was 1.0%. The particles are prone to be partially agglomerated at the high concentration of WSPALR and calcined WSPALR (Figure 2), therefore, the elongation at break of the coating decreased.

This answer is incorporated in the text on page 3, line 142-144, page 4, line 181-185 and page 11, line 335-346 in the revised version, thank you very much.

Q9: Were the coated wood samples swelling in moist condition like mould resistance test? If so, coating might break due to reaching an ultimate tensile strength.

A: The coated wood samples hardly swell and the swelling can be neglected in mould resistance test.

This answer is incorporated in the text on page 5, line 191 in the revised version, thank you very much.

Q10: How much can the waterborne coating prevent wood from swelling?

A: The waterborne coatings can make wood expansion rate less than 0.5%.

This answer is incorporated in the text on page 5, line 190 in the revised version, thank you very much.

Q11: Where the Trichoderma came from? inside poplar wood?

A: Trichoderma came from the poplar veneers. The coating itself did not grow mildew.

This answer is incorporated in the text on page 14, line 390 in the revised version, thank you very much.

Q12: In Figure 8 I could not recognize the Trichoderma generate portion. Did it grow between coating and wood? Please point out the place.

A: Trichoderma generate portion was between the coating and the wood (as shown in the red box in Figure 11(A) and (B)).

This answer is incorporated in the text on page 14, line 393-394 in the revised version, thank you very much.

Q13: To conclude an optimum condition for something, there must be extremely narrow state of interest. Do you think your optimum concentration of 0.5% for WSPALR calcined at high temperature in coating can stand for another preparation of coating methods?

A: According to the experiment, the concentration of 0.5% for WSPALR calcined at high temperature in coating is a better concentration, which can stand for another preparation of coating methods. The term “optimum” has been changed to “better”.

This answer is incorporated in the text on page 16, line 425-427 in the revised version, thank you very much.

Q14: It must be careful for use the term “optimum”.

A: The term “optimum” has been changed to “better”.

This answer is incorporated in the text on page 16, line 425 in the revised version, thank you very much.

Q15: An English proof reading by a native speaker should be recommended because there are some misspellings, strange and confused sentences and words.

A: English has been checked by a native speaker.

PS: All the revised parts in the manuscript were highlighted by green color.

Round  3

Reviewer 3 Report

Your additional experiments showed better understanding in some parts of your manuscript. However it must be careful for measuring mechanical properties of thin film especially elongation. How did you measure the elongation at break or fracture strain ? Did you use non-contact displacement gauge like an optical displacement meter ?

I did not know why the fracture strain increased by adding filler, it usually becomes brittle otherwise increased strength obtained. Gripping of thin film arises problems like slippage or fracture of a film at chucking, those lead to unstable tensile tests.

If you can explain this phenomenon referring some other works, you can add the elongation results. If not, it is not good for showing the elongation results. How about strength results?

Author Response

The reviewer gives very valuable comments on our manuscript, and we would like to take this opportunity to express our great appreciation on him/her as well as the comments. Followings are our responses to the reviewer’s questions and comments together with the changes made in the revised version.

Q1: Your additional experiments showed better understanding in some parts of your manuscript. However it must be careful for measuring mechanical properties of thin film especially elongation. How did you measure the elongation at break or fracture strain? Did you use non-contact displacement gauge like an optical displacement meter?

A: The coatings were prepared into thin strips, and then the two ends of the coatings were fixed with fixtures. At a certain tension speed (0.12mm/min), the coating was deformed until it was destroyed by applying load on the longitudinal direction of the coatings. The elongation at break of the coatings was tested by the Model AG-IC100KN precision electronic universal capability experiment machine (Shimadzu Co., Ltd., Kyoto, Japan) and TRview X optical displacement meter.

This answer is incorporated in the text on page 4, line 183, and on page 5, 186-188 in the revised version, thank you very much.

Q2: I did not know why the fracture strain increased by adding filler, it usually becomes brittle otherwise increased strength obtained. Gripping of thin film arises problems like slippage or fracture of a film at chucking, those lead to unstable tensile tests.

A: When the concentration of WSPALR increased from 0 to 1.0%, the elongation at break increased from 2.6% to 5.1% for a reinforcement with cellulose fibers leading to significantly higher values of fracture strain [39]. When the concentration of high-temperature calcined WSPALR increased from 0 to 1.0%, the elongation at break increased from 2.6% to 4.8% for the improved toughening induced by the ash particles (ash comes from the calcined WSPALR) [40]. During the tensile test, the two ends of the coating sample were fixed firmly to ensure that the coating did not slide. In the tensile test, the crack was not in the coating chuck, but in the middle of the coating. Tensile test was repeated at least four times with an error of less than 5.0%.

This answer is incorporated in the text on page 5, line 188-190 and on page 11, line 343-344, line 347-348 in the revised version, thank you very much.

Q3: If you can explain this phenomenon referring some other works, you can add the elongation results. If not, it is not good for showing the elongation results. How about strength results?

A: The increased fracture strain by adding filler has been explained in references [39] and [40], so the experimental results have been added. Tensile strength is measured without a base material, which can not reflect the strength of wood surface coating, and it has little significance for the strength of wood coatings. Therefore, the tensile strength is not shown. However, the impact strength (resistance to impact) of the waterborne wood coatings has been shown in Figure 8.

[39] Zarges, J.C.; Minkley, D.; Feldmann, M.; Heim, H.P. Fracture toughness of injection molded, man-made cellulose fiber reinforced polypropylene. Compos. Part A-Appl. S. 2017, 98, 147–158.

[40] Igarza, E.; Pardo, S.G.; Abad, M.J.; Cano, J.; Galante, M.J.; Pettarin, V.; Bernal, C. Structure-fracture properties relationship for polypropylene reinforced with fly ash with and without maleic anhydride functionalized isotactic polypropylene as coupling agent. Mater. Design 2014, 55, 85–92.

This answer is incorporated in the text on page 11, line 343-344, line 347-348 and page 16, line 429 in the revised version, thank you very much.

PS: All the revised parts in the manuscript were highlighted by red color.

This manuscript is a resubmission of an earlier submission. The following is a list of the peer review reports and author responses from that submission.

Round  1

Reviewer 1 Report

A brief summary

The authors investigated the influence of the addition of cellulose nanocrystals (CNC) and of calcined CNCs on some properties of waterborne wood coating (hardness, adhesion, resistance to impact, colour and resistance to moulds. In general, the selected properties were improved, but not substantially influencing on colour of the coating.

Broad, general comments

The quality of written English language is relatively good, but nevertheless it should be improved. I suggest the assistance of a professional language editor or of a native English speaker.

Specific comments

L 12-26:            The quality of English written language could be improved, not so much from the grammatical point of view but more in terms of the writing style.

L 13:                 Instead of the term “impact strength” I suggest the authors to use the term “resistance to impact”. Instead the “mildew” the “mould” or “mold” should be used. These two remarks should be followed in the complete paper.

L 27:                 The order of the keywords (and some new ones) should be: waterborne coatings, cellulose nanocrystals (CNC); calcined CNC; mechanical properties; resistance to moulds 

L. 37-38:           Grammatical error: “The most important feature of waterborne coatings is that it does not contain volatile organic compounds and air pollutants [5].” Should be written as “The most important feature of waterborne coatings is that they do not contain volatile organic compounds and air pollutants [5].”

L 39-43:            “Wood is rich in various nutrients such as sugar and starch for microbial growth, and wood extracts lack bactericidal substances to inhibit microbial growth, therefore, microbial growth easily leads to wood mildew and discoloration, reduces the wood grade and use value, and is not conducive to subsequent processing and utilization [7].” I have several comments here. Plenty of wood extracts exhibit biocidal activity (dependent on the wood species) and in fact inhibit microbial growth. But there is the question if the extent of such protection is sufficient. Secondly, decay by bacteria is the least important, so I would not talk here about bactericidal substances. I suggest reformulation of the first part of the sentence into something like “Wood is rich in various nutrients such as sugar and starch for microbial growth, and therefore it is not resistant to fungal and bacterial infestations…”or something similar (like “it is susceptible to fungal and bacterial infestations…). Secondly, in the field of wood science & technology (including protection” the term mildew is less frequently used. I suggest to use the term mould instead. (“Mould” is written in British English and in American English it could be written “mold”) This remark should be followed in the whole paper. Thirdly, I do not understand what is meant by “wood grade”. Please rewrite this part of the sentence to make it more clear.

L 43:                 Instead of “When waterborne coatings are coated on the wood surface…” say “When waterborne coatings are applied on the wood surface…”

L44-45:             In “…surface become green, black” change into “…makes the surface become discoloured, blue, gree, or black and…”. “…and even make the film fall off…” Is this really true? Usually this does not happen. I suggest that you write in a more “mild” form: “..and may even decrease the adhesion strength of the film…”

L 53:                 Change “mildew” into “mold”. And also in the next sentence in the Line 54. At this point I stop to suggest that you change “mildew” into “mold”. Please change wherever relevant in the paper.

L79:                  Change “…, mix evenly” into “…, mixed evenly”.

L84:                  “The viscosity of the coating is 20s”. Please add a short explanation, how this viscosity was measured – (like with the Ford cup No. 4 in accordance with the standard XY..) or something similar

L90:                  Change “..was sifted through 250 μm.” into “..was sieved through 250 μm mesh.” In the sentence in L91, change “sifting” into “sieving”

L93:                  This sounds strange: “was packed in a beaker” Write “was poured into a beaker” or something similar

L 101-102:       “The putty was coated evenly on the poplar veneers. When the putty was dry, the poplar veneers were sanded using 1000 grit sandpaper and a dry cloth was used to wipe off the dust.” Now, this demands some additional explanations and discussion”. Please write/explain elsewhere in the paper (where it is relevant – might be already in the introduction, here and also in the discussion (especially in connection with the discussion about the adhesion strength) why actually did you use potty? This is quite unusual. And especially, why did your perform sanding with such a high grit? This for sure made the surface very smooth. By this way, you prevented mechanical anchoring of the coating into wood pores, but also mechanical anchoring of the coating into very smooth putty surface very likely became impossible. This could have influence to a large extent to the measured values of the adhesion strength. So, please explain this issue in the paper!

L 104-105:       also intermediate sanding was performed with unusually high paper grit? Why? Usually we perform intermediate sanding with maximal grit of about 400-600

L101-114:        Please explain why the drying time after application of the first layer of the finish without CNC was30 minutes and in the case of coatings with the addition of CNC and calcined CNC was about 3 hours? Why this difference?

L 132:               “The hardness of pencils (6H, 5H, 4H, 3H, 2H, 1H, HB, 1B, 2B, 3B, 4B, 5B and 6B) was..”: you probably meant “The hardness of films (determined with the 6H, 5H, 4H, 3H, 2H, 1H, HB, 1B, 2B, 3B, 4B, 5B and 6B pencils) was..”

L140:                change “…..of CNC were shown in Figure 1..” into “…..of CNC are shown in Figure 1..”

L145:                “…which were composites of CNC and some silicon..”. I do not understand this quite well. Why “composites”? What kind of? Silicon in which form/compound ? Please add some explanations, or rewrite in such a way to avoid ambiguity.

L151-152:        Instead of “(E) high-temperature calcination of CNC” write “(E) high-temperature calcined CNC”

L153-163:        (a) FT-IR terminology: in the case of FT-IR spectra we usually do not talk about the peaks but rather of the bands. So, for instance the expression “The absorption peak at 3350 cm-1 is –OH stretching vibration..” should be changed into “The absorption band at 3350 cm-1 is assigned –OH stretching vibration..” Make similar changes everywhere in the paper, related to FT-IR spectra; (b) references: please add the references for the assignments of various bands mentioned (c) contents: I do not really understand the last sentence. What kind of ash do you have in minds? Was there any ash on the surface of wheat straw powder? And why would the intensities of the mentioned bands be increased, if the “ash” is removed? Please, add explanation, rewrite to make this part more clear, etc.

L 169-170:       Change “The effect of CNC before and after high-temperature  calcination on the hardness of waterborne coatings was shown in Figure 3.” into “The effect of CNC before and after high-temperature  calcination on the hardness of waterborne coatings is shown in Figure 3.”

L 167-185:       Do you know, or think of any reasons that it might be possible for the both curves in Fig. 3 have maxima – this means, why in both cases the hardness was the highest when 1-2 % or 0.5-1% of CNC or calcined CNC was added and not in the case of the highest concentrations of the added CNCs?

L 183:               I am not familiar with the expression “CNC carbided”. Is it properly used in this case? Please check and change if it turns out that the expression “carbided” is not the best or the relevant one.

L 188-207:       In this part of the paper (adhesion) it should be taken into account that you are not talking about the adhesion on poplar wood, but on putty treated poplar wood or most probably on putty and not on wood (I do not know what was the thickness of the putty layer on wood substrate). Please discuss this issue in this part of the discussion. And as already said, the smoothness was most probably high or even too high (sanding with 1000 grit!). Please have in minds this fact in your discussion as well, make some comments about this, etc.

L 210:               change “Impact resistance referred..” into “Impact resistance referrs..” And similarly in the next sentence in the Line 211 change “showed” into “shows”

L 210-224:       Similarly as above, I am asking if you have any idea, why the curves in Fig. 5 have maxima (although small), that means why with the increase of both CNC concentration after the optimal concentration of CNC impact resistance (at least in the case of blue line) decreased. Try to discuss this issue, if possible. Also, try to correlate in the discussion the impact strength resistance with the hardness. It seems that both series of the results are related. Is this expected or not, the reasons, etc…

L232-233:        change “respectively, are expressed as illumination difference, red-green index difference, and yellow-blue index difference” into “respectively, are expressed as the lightness difference, red-green colour difference, and yellow-blue colour difference”

L 243:               change “waterborne coatings was shown in Table 3.” Into “waterborne coatings is shown in Table 3.”

L 244:               “From the data in Table 1, When CNC was not added, the color difference of the coating was very small and uniform.”. This is not understandable. Why at all should the colour change, if nothing was added? Because of some exposure, keeping the samples (for what time) somewhere in the lab, or for some other reason? Please provide explanation? Or is the calculated Delta E value just the result of experimental errors during measurements?

L 245-247:       “When the content of CNC was 5.0%, the color difference of the coating increased significantly, reaching the maximum value of 8.4. This is because when the content of CNC was higher, it was not easy to disperse and formed reunion.” What about the colours of CNC and calcined CNC. Especially the calcined one. Was it black? The addition could also influence on colour because of the colour of the additive (CNC)? And how do you know, that the CNC was not easily dispersed and formed reunion? Please explain! And also, what does it mean “to form reunion”? Did you maybe think of compatibility / coupling between the matrix (the coating) and CNC?

Table 3: where this acceptability data are taken from? Please cite the reference!

L 251-253:       “This is because when the content of high-temperature calcined CNC was higher, it was easy to aggregate and not disperse uniformly.” How do you know this?

L 259:               Change “The results of mould resistance of the coating were shown in Figure 6.” Into “The results of mould resistance of the coating are shown in Figure 6.”

L 259-271:       The quality of English written language is a bit lower that in the other parts of the paper, so it is hard to read and understand it. Please correct.

L 263-271:       How do you know that the inhibition effect has the reason in “CNC itself was long rod-like and accompanied by some tubular substances”? please explain, maybe support by some similar data/reports from literature (references). The same relates also to “micro-pores” as a possible reason.

L 273:               Change “growed” into “grew”

L 277-292:       Also in the “Conclusion” the quality of Englsih written language should be improved.

Author Response

Q1: The authors investigated the influence of the addition of cellulose nanocrystals (CNC) and of calcined CNCs on some properties of waterborne wood coating (hardness, adhesion, resistance to impact, colour and resistance to moulds. In general, the selected properties were improved, but not substantially influencing on color of the coating.

A: In this paper, wheat straw powder after lignin removal (WSPALR) was obtained from wheat straw treated with alkaline hydrogen peroxide. The effect of WSPALR on the mould resistance of waterborne coatings before and after high-temperature calcination was studied, and the color difference, hardness, adhesion, and impact resistance of waterborne coatings were also studied, from which the properties of waterborne coatings were analyzed to improve the mould resistance of waterborne coatings while maintaining their original properties.

Q2: The quality of written English language is relatively good, but nevertheless it should be improved. I suggest the assistance of a professional language editor or of a native English speaker.

A: A native English speaker helped us to improve the quality of English.

Q3: L 12-26: The quality of English written language could be improved, not so much from the grammatical point of view but more in terms of the writing style.

A: A native English speaker helped us to improve the quality of English.

Q4: L 13: Instead of the term “impact strength” I suggest the authors to use the term “resistance to impact”. Instead the “mildew” the “mould” or “mold” should be used. These two remarks should be followed in the complete paper.

A: The term “impact strength” has been changed into “resistance to impact”. The term “mould” has been used in the whole paper.

Q5: L 27: The order of the keywords (and some new ones) should be: waterborne coatings, cellulose nanocrystals (CNC); calcined CNC; mechanical properties; resistance to moulds.

A: The order of the keywords (and some new ones) has been added.

Q6: L. 37-38: Grammatical error: “The most important feature of waterborne coatings is that it does not contain volatile organic compounds and air pollutants [5].” Should be written as “The most important feature of waterborne coatings is that they do not contain volatile organic compounds and air pollutants [5].”

A: The grammatical error has been corrected.

Q7: L 39-43: “Wood is rich in various nutrients such as sugar and starch for microbial growth, and wood extracts lack bactericidal substances to inhibit microbial growth, therefore, microbial growth easily leads to wood mildew and discoloration, reduces the wood grade and use value, and is not conducive to subsequent processing and utilization [7].” I have several comments here. Plenty of wood extracts exhibit biocidal activity (dependent on the wood species) and in fact inhibit microbial growth. But there is the question if the extent of such protection is sufficient. Secondly, decay by bacteria is the least important, so I would not talk here about bactericidal substances. I suggest reformulation of the first part of the sentence into something like “Wood is rich in various nutrients such as sugar and starch for microbial growth, and therefore it is not resistant to fungal and bacterial infestations…”or something similar (like “it is susceptible to fungal and bacterial infestations…). Secondly, in the field of wood science & technology (including protection” the term mildew is less frequently used. I suggest to use the term mould instead. (“Mould” is written in British English and in American English it could be written “mold”) This remark should be followed in the whole paper. Thirdly, I do not understand what is meant by “wood grade”. Please rewrite this part of the sentence to make it more clear.

A: The sentence has been changed into “Wood is rich in various nutrients such as sugar and starch for microbial growth, and therefore it is not resistant to fungal and bacterial infestations, which reduce the wood quality and use value, and make it not conducive to subsequent processing and utilization.” The word “mould” has been used in the whole paper. The word “wood grade” has been changed to “wood quality”.

Q8: L 43: Instead of “When waterborne coatings are coated on the wood surface…” say “When waterborne coatings are applied on the wood surface…”.

A: The word “coated” has been changed into “applied”.

Q9: L44-45: In “…surface become green, black” change into “…makes the surface become discoloured, blue, green, or black and…”. “…and even make the film fall off…” Is this really true? Usually this does not happen. I suggest that you write in a more “mild” form: “..and may even decrease the adhesion strength of the film…”

A: The sentence has been changed into “mould may cause discoloration and even decrease the adhesion strength of the film”.

Q10: L 53: Change “mildew” into “mold”. And also in the next sentence in the Line 54. At this point I stop to suggest that you change “mildew” into “mold”. Please change wherever relevant in the paper.

A: The term “mildew” has been changed into “mould” in the whole paper.

Q11: L79: Change “…, mix evenly” into “…, mixed evenly”.

A: “…, mix evenly” has been changed into “…, mixed evenly”.

Q12: L84: “The viscosity of the coating is 20s”. Please add a short explanation, how this viscosity was measured – (like with the Ford cup No. 4 in accordance with the standard XY..) or something similar

A: The capacity of the Ford cup No. 4 is 100 mL. When the container is full of the waterborne wood coating, the time it takes to flow out from the standard bottom hole is used to measure the coating viscosity in seconds (s) [26].

Q13: L90: Change “..was sifted through 250 μm.” into “..was sieved through 250 μm mesh.” In the sentence in L91, change “sifting” into “sieving”.

A: “..was sifted through 250 μm.” has been changed into “..was sieved through 250 μm mesh.” The word “sifting” has been changed into “sieving”.

Q14: L93: This sounds strange: “was packed in a beaker” Write “was poured into a beaker” or something similar.

A: “was packed in a beaker” has been changed into “was poured into a beaker”.

Q15: L 101-102: “The putty was coated evenly on the poplar veneers. When the putty was dry, the poplar veneers were sanded using 1000 grit sandpaper and a dry cloth was used to wipe off the dust.” Now, this demands some additional explanations and discussion. Please write/explain elsewhere in the paper (where it is relevant – might be already in the introduction, here and also in the discussion (especially in connection with the discussion about the adhesion strength) why actually did you use putty? This is quite unusual. And especially, why did your perform sanding with such a high grit? This for sure made the surface very smooth. By this way, you prevented mechanical anchoring of the coating into wood pores, but also mechanical anchoring of the coating into very smooth putty surface very likely became impossible. This could have influence to a large extent to the measured values of the adhesion strength. So, please explain this issue in the paper!

A: The main ingredients of putty are talc powder and water, which are used to fill the vessels of poplar wood. The sandpaper used for putty is 600 grit, and the sandpaper used for primer and topcoat painting is 1000 grit. The sentence has been changed into “The putty was coated evenly on the poplar as a veneer. When the putty was dry, the poplar veneer was sanded using 600 grit sandpaper, and a dry cloth was used to wipe off the dust.”

Q16: L 104-105:  also intermediate sanding was performed with unusually high paper grit? Why? Usually we perform intermediate sanding with maximal grit of about 400-600.

A: The sandpaper used for primer and topcoat painting is 1000 grit. Because waterborne wood coatings are smooth on the surface of wood and therefore polished with high grit sandpaper.

Q17: L101-114: Please explain why the drying time after application of the first layer of the finish without CNC was30 minutes and in the case of coatings with the addition of CNC and calcined CNC was about 3 hours? Why this difference?

A: Beacause the addition of WSPALR and calcined WSPALR increased the viscosity of the coating and resulted in slow volatilization of solvents and long drying time.

Q18: L 132: “The hardness of pencils (6H, 5H, 4H, 3H, 2H, 1H, HB, 1B, 2B, 3B, 4B, 5B and 6B) was..”: you probably meant “The hardness of films (determined with the 6H, 5H, 4H, 3H, 2H, 1H, HB, 1B, 2B, 3B, 4B, 5B and 6B pencils) was..”

A: The sentence has been changed into “The hardness of films (determined with the 6H, 5H, 4H, 3H, 2H, 1H, HB, 1B, 2B, 3B, 4B, 5B and 6B pencils) was..”

Q19: L140: change “…..of CNC were shown in Figure 1..” into “…..of CNC are shown in Figure 1..”

A: The word “were shown” has been changed into “are shown”.

Q20: L145: “…which were composites of CNC and some silicon..”. I do not understand this quite well. Why “composites”? What kind of? Silicon in which form/compound? Please add some explanations, or rewrite in such a way to avoid ambiguity.

A: The sentence has been changed into “…which were cellulose and SiO2 (Si comes from ash in wheat straw powder).”

Q21: L151-152: Instead of “(E) high-temperature calcination of CNC” write “(E) high-temperature calcined CNC”

A: The word “calcination of” has been changed into “calcined”.

Q22: L153-163:  (a) FT-IR terminology: in the case of FT-IR spectra we usually do not talk about the peaks but rather of the bands. So, for instance the expression “The absorption peak at 3350 cm-1 is –OH stretching vibration..” should be changed into “The absorption band at 3350 cm-1 is assigned –OH stretching vibration.” Make similar changes everywhere in the paper, related to FT-IR spectra; (b) references: please add the references for the assignments of various bands mentioned (c) contents: I do not really understand the last sentence. What kind of ash do you have in minds? Was there any ash on the surface of wheat straw powder? And why would the intensities of the mentioned bands be increased, if the “ash” is removed? Please, add explanation, rewrite to make this part more clear, etc.

A: (a) The “peak” has been changed into “band”; (b) The references [30-34] have been added; (c) Compared with untreated wheat straw powder, the C–O stretching vibration band and the C–C skeleton stretching vibration absorption at 1160 cm-1 increased after the alkaline hydrogen peroxide, which indicated that the purity of SiO2 and cellulose was improved because the removal of lignin leaves a large amount of cellulose and SiO2 after the reaction.

Q23: L 169-170: Change “The effect of CNC before and after high-temperature calcination on the hardness of waterborne coatings was shown in Figure 3.” into “The effect of CNC before and after high-temperature calcination on the hardness of waterborne coatings is shown in Figure 3.”

A: The word “was” has been changed into “is”.

Q24: L 167-185: Do you know, or think of any reasons that it might be possible for the both curves in Fig. 3 have maxima – this means, why in both cases the hardness was the highest when 1-2 % or 0.5-1% of CNC or calcined CNC was added and not in the case of the highest concentrations of the added CNCs?

A: The agglomeration of particles easily occurs due to the large amount of WSPALR and calcined WSPALR (Figure 2). The distribution is uneven, leading to a decrease of hardness [35].

Q25: L 183: I am not familiar with the expression “CNC carbided”. Is it properly used in this case? Please check and change if it turns out that the expression “carbided” is not the best or the relevant one.

A: The word “carbided” has been changed into “decomposed”.

Q26: L 188-207: In this part of the paper (adhesion) it should be taken into account that you are not talking about the adhesion on poplar wood, but on putty treated poplar wood or most probably on putty and not on wood (I do not know what was the thickness of the putty layer on wood substrate). Please discuss this issue in this part of the discussion. And as already said, the smoothness was most probably high or even too high (sanding with 1000 grit!). Please have in minds this fact in your discussion as well, make some comments about this, etc.

A: The adhesion of waterborne coatings refers to the ability to form strong bonding between coatings and poplar substrates. Putty was used to fill the vessels of poplar wood and did not increase coating thickness.

Q27: L 210: change “Impact resistance referred..” into “Impact resistance refers.” And similarly in the next sentence in the Line 211 change “showed” into “shows”

A: The word “referred” has been changed into “refers”. The word “showed” has been changed into “shows”.

Q28: L 210-224: Similarly as above, I am asking if you have any idea, why the curves in Fig. 5 have maxima (although small), that means why with the increase of both CNC concentration after the optimal concentration of CNC impact resistance (at least in the case of blue line) decreased. Try to discuss this issue, if possible. Also, try to correlate in the discussion the impact strength resistance with the hardness. It seems that both series of the results are related. Is this expected or not, the reasons, etc…

A: The particles are prone to agglomeration at the high concentration of WSPALR and calcined WSPALR (Figure 2), which will cause uneven distribution and fluctuation of resistance to impact [35]. The resistance to impact of the waterborne coatings was basically unchanged with an increase of WSPALR concentration from 0.5% to 5.0%, which indicated that the resistance to impact of the waterborne wood coatings could be increased by adding WSPALR before and after high-temperature calcination because they can absorb more impact energy [36].

Q29: L232-233: change “respectively, are expressed as illumination difference, red-green index difference, and yellow-blue index difference” into “respectively, are expressed as the lightness difference, red-green colour difference, and yellow-blue colour difference”.

A: The sentence has been changed.

Q30: L 243: change “waterborne coatings was shown in Table 3.” Into “waterborne coatings is shown in Table 3.”

A: The word “was” has been changed into “is”.

Q31: L 244: “From the data in Table 1, when CNC was not added, the color difference of the coating was very small and uniform.”. This is not understandable. Why at all should the colour change, if nothing was added? Because of some exposure, keeping the samples (for what time) somewhere in the lab, or for some other reason? Please provide explanation? Or is the calculated Delta E value just the result of experimental errors during measurements?

A: The coating itself is the composite, and the components may be unevenly distributed, so the formation of the coating film has color difference.

Q32: L 245-247: “When the content of CNC was 5.0%, the color difference of the coating increased significantly, reaching the maximum value of 8.4. This is because when the content of CNC was higher, it was not easy to disperse and formed reunion.” What about the colours of CNC and calcined CNC. Especially the calcined one. Was it black? The addition could also influence on colour because of the colour of the additive (CNC)? And how do you know, that the CNC was not easily dispersed and formed reunion? Please explain! And also, what does it mean “to form reunion”? Did you maybe think of compatibility / coupling between the matrix (the coating) and CNC?

A: The color of WSPALR is yellow and the color of calcined WSPALR is red. WSPALR and calcined WSPALR have color, which may also affect the color difference. When the concentration of WSPALR was 5.0%, the color difference of the coating increased significantly, reaching the maximum value of 8.4. This is because when the concentration of WSPALR was higher, it was not easy to disperse (Figure 2) due to the compatibility between the coating and WSPALR.

Q33:Table 3: where this acceptability data are taken from? Please cite the reference!

A: The reference [37] has been cited.

Q34: L 251-253: “This is because when the content of high-temperature calcined CNC was higher, it was easy to aggregate and not disperse uniformly.” How do you know this?

A: Figure 2 has been added and showed that when the concentration of high-temperature calcined WSPALR was higher, it easily aggregated and did not disperse uniformly.

Q35: L 259: Change “The results of mould resistance of the coating were shown in Figure 6.” Into “The results of mould resistance of the coating are shown in Figure 6.”

A: The word “were” has been changed into “are”.

Q36: L 259-271: The quality of English written language is a bit lower that in the other parts of the paper, so it is hard to read and understand it. Please correct.

A: A native English speaker helped us to improve the quality of English.

Q37: L 263-271: How do you know that the inhibition effect has the reason in “CNC itself was long rod-like and accompanied by some tubular substances”? please explain, maybe support by some similar data/reports from literature (references). The same relates also to “micro-pores” as a possible reason.

A: WSPALR was long, rod-like, and accompanied by some tubular substances (Figure 1(C)). When WSPALR was calcined at high temperature, carbon was basically transformed into carbon dioxide and escaped to form porous and irregular composites with micropores (Figure 1(E)), which had activity and can prevent the propagation of fungi [38], so that the inhibition effect of high-temperature calcined WSPALR on Trichoderma was better than that of WSPALR.

Q38: L 273: Change “growed” into “grew”.

A: The “growed” has been changed into “grew”.

Q39: L 277-292: Also in the “Conclusion” the quality of Englsih written language should be improved.

A: A native English speaker helped us to improve the quality of English.

Reviewer 2 Report

The paper by Yan et al. purportedly presents the obtaining of waterborne coatings on wood surfaces.

In my opinion, the paper can not be accepted for publication in Coatings due to the following reasons:

1.       Cellulose nanocrystals use in wood impregnation would be interesting to study, but as the authors designed the study, calcining the cellulose nanocrystals at the mentioned temperatures leads mainly to ash. No reason for obtaining the CNCs in the first place. So the whole idea and design of this paper is flawed.

2.       The introduction part lacks the description of the novelty of this study.

3.       The materials and methods section do not describe in detail the methodology for impregnants obtaining and coating of the wood.

4.       Results and discussion: actually it is the impregnated poplar wood samples that needed the SEM morphology discussion, not so much the raw materials, as it was presented in figure 1,

5.       The FTIR spectroscopy results are not discussed in enough detail, and again, wood+impregnant system should have been discussed here;

6.       The mechanism underlying CNC influence on mechanical properties is not discussed, only descriptive information is given

7.       The anti-mold testing methodology is not clearly discussed. This applies also to the discussion part about this issue. Figure 6 does not have sufficient resolution to discern between the samples.

Author Response

Q1: Cellulose nanocrystals use in wood impregnation would be interesting to study, but as the authors designed the study, calcining the cellulose nanocrystals at the mentioned temperatures leads mainly to ash. No reason for obtaining the CNCs in the first place. So the whole idea and design of this paper is flawed.

A: In this paper, wheat straw powder after lignin removal (WSPALR) was obtained from wheat straw treated with alkaline hydrogen peroxide. The effect of WSPALR on the mould resistance of waterborne coatings before and after high-temperature calcination was studied, and the color difference, hardness, adhesion, and impact resistance of waterborne coatings were also studied, from which the properties of waterborne coatings were analyzed to improve the mould resistance of waterborne coatings while maintaining their original properties.

Q2: The introduction part lacks the description of the novelty of this study.

A:The description of the novelty has been added in the introduction. The cellulose nanocrystals are not easy to prepare and obtain, and they are very expensive. If filamentous substances can be obtained by a simple method and have good mould resistance, they will have good prospects. Wheat straw is the residual part of wheat left after harvesting the seeds. It is a kind of renewable biological resource with many uses, high fiber content, and lignin. Wheat straw is rich in potassium, calcium, magnesium, and organic matter. In this paper, wheat straw powder after lignin removal (WSPALR) was obtained from wheat straw treated with alkaline hydrogen peroxide. The effect of WSPALR on the mould resistance of waterborne coatings before and after high-temperature calcination was studied, and the color difference, hardness, adhesion, and impact resistance of waterborne coatings were also studied, from which the properties of waterborne coatings were analyzed to improve the mould resistance of waterborne coatings while maintaining their original properties.

Q3: The materials and methods section do not describe in detail the methodology for impregnants obtaining and coating of the wood.

A: The methodology for obtaining the coating has been described in detail. The main ingredients of putty are talc powder and water, which are used to fill the vessels of poplar wood. The putty was coated evenly on the poplar as a veneer. When the putty was dry, the poplar veneer was sanded using 600 grit sandpaper, and a dry cloth was used to wipe off the dust. The waterborne wood coating was sprayed using an airbrush (Guangzhou Zhongtian Electrical Equipment Co. Ltd., Guangzhou, China) on poplar veneers, waiting 30 min for natural drying, and was sanded using 1000 grit sandpaper; then a dry cloth was used to wipe off the dust. The spray process of waterborne wood coating needed to be repeated twice. WSPALR was added to waterborne wood coatings at 0, 0.5%, 1.0%, 2.0%, 3.5%, and 5.0% concentrations, and mixed evenly. The waterborne coatings with different concentrations of WSPALR were sprayed separately using an airbrush. The coating was naturally dried for 3 h and then sanded using 1000 grit sandpaper, and a dry cloth was used to wipe off the dust. The above process was repeated twice. Similarly, waterborne coatings with 0, 0.5%, 1.0%, 2.0%, 3.5%, and 5.0% concentrations of calcined WSPALR were also prepared. The preparation method of waterborne coating with the addition of WSPALR after calcination at high temperature was the same as that described above. The thickness of the dry waterborne coating was about 40 µm.

Q4: Results and discussion: actually it is the impregnated poplar wood samples that needed the SEM morphology discussion, not so much the raw materials, as it was presented in figure 1.

A: The SEM image of the waterborne coatings with WSPALR and calcined WSPALR added are shown in Figure 2, indicating that the agglomeration of WSPALR or calcined WSPALR easily occurs due to the large numbers of particles in the waterborne coatings (Figure 2(C) and Figure 2(E)).

Q5: The FTIR spectroscopy results are not discussed in enough detail, and again, wood+impregnant system should have been discussed here

A: FTIR spectrum results have been discussed in detail. The infrared spectrum of the waterborne coatings added with WSPALR and calcined WSPALR are basically unchanged (Figure 4), indicating that WSPALR or calcined WSPALR and waterborne coatings are physically bonded.

Q6: The mechanism underlying CNC influence on mechanical properties is not discussed, only descriptive information is given.

A: The mechanism underlying WSPALR influence on mechanical properties has been discussed. The agglomeration of particles easily occurs due to the large amount of WSPALR and calcined WSPALR (Figure 2). The distribution is uneven, leading to a decrease of hardness [35]. The hardness of waterborne coatings can be improved by adding WSPALR before and after high-temperature calcination, and the hardness of waterborne coatings with WSPALR before high-temperature calcination was higher than that after high-temperature calcination. This is because after calcination of WSPALR at high temperature, WSPALR decomposed and changed its structural morphology from long rods to a porous structure with micropores, and the multi-void structure was not conducive to the improvement of hardness of the coating.

From the results shown in Figure 6, when WSPALR was added before and after calcination at high temperature at the concentration of 0.5%, the adhesion of waterborne coating was better, which was level 1. After sodium hydroxide treatment, lignin was removed on the surface of the wheat straw powder, and also certain hemicellulose and other substances were removed, which was not conducive to enhancing interface bonding with waterborne coatings, and when the concentration of WSPALR and high-temperature calcined WSPALR was higher, it easily aggregated and did not disperse uniformly, so adding a certain amount of WSPALR before and after high-temperature calcination decreased the adhesion of waterborne coatings.

The data in Figure 7 indicates that its overall fluctuation range was very small. The particles are prone to agglomeration at the high concentration of WSPALR and calcined WSPALR (Figure 2), which will cause uneven distribution and fluctuation of impact resistance [35]. The resistance to impact of the waterborne coatings was basically unchanged with an increase of WSPALR concentration from 0.5% to 5.0%, which indicated that the resistance to impact of the waterborne wood coatings could be increased by adding WSPALR before and after high-temperature calcination because they can absorb more impact energy [36].

Q7: The anti-mould testing methodology is not clearly discussed. This applies also to the discussion part about this issue. Figure 6 does not have sufficient resolution to discern between the samples.

A: Mould resistance was tested by the suspension method according to GB/T 1741-2007 [29]. The samples were placed in an incubator. The temperature was between 25°C and 30°C and relative humidity was 85.0%. After 5 d, mould growth on the surface was observed. The sufficient resolution to discern between the samples has been improved (Figure 8). The more WSPALR was added, the better the inhibition effect on Trichoderma. This is because WSPALR itself is long and rod-like and accompanied by some tubular substances (Figure 1(C)), which had an inactivating effect on Trichoderma and afforded a certain degree of mould resistance. When WSPALR was calcined at high temperature, carbon was basically transformed into carbon dioxide and escaped to form porous and irregular composites with micropores (Figure 1(E)), which had activity and can prevent the propagation of fungi [38], so that the inhibition effect of high-temperature calcined WSPALR on Trichoderma was better than that of WSPALR.

Reviewer 3 Report

General Remarks

Althought I find the topic very interesting, the puplication has several methological flaws that have to be adressed prior to publication.

My biggest remark is that nanocrystalline cellulose is mentioned in the title. Nanocrystalline cellulose is usually prepared by acid hydrolysis. The authors however describe a typical bleaching process instead.

Detailed Remarks:

- Could the authors replace the term "mildew" with the more common term "mould growth".

Materials and Methods:

- Could the authors add the number of samples, wood species (botanical name, e.g. Poplus nigra) and surface preparation (sanding, planing) in the material section

- The number of samples/number of replicate measurements are missing

- I am missing the method for the formation of "real" nanocrystalline cellulose - the described method refers only to delignification.

- The SEM images are not convincing evidence of the presence of nanocellulose, since the magnification is way too low.

- Could the authors explain, what purpose the putty serves?

- Please indicate the dry film thickness of the coating, since this has a major impact on the mechanical properties of the coatings

- Could the authors explain briefly, how the assessment of the impact test works: how is the impact force varied?

Results and Discussion

- Could the authors explain, where the Si is coming from? I

- Surface hardness is one of many performance criteria of coatings and depends on the application. Hence, harder surfaces are not always better than softer ones.

- Figure 3 und Figure 4 are not very convincing. Firstly, is it true that a coating with no CNC has a hardness of 1/adhesion of 0/Impact strength of 0 kg cm?.Secondly, are the differences of the datapoints significant?. I am missing some sort of measurement of spread.

Author Response

Q1: My biggest remark is that nanocrystalline cellulose is mentioned in the title. Nanocrystalline cellulose is usually prepared by acid hydrolysis. The authors however describe a typical bleaching process instead.

A: Nanocrystalline cellulose has been deleted.

Q2: Could the authors replace the term "mildew" with the more common term "mould growth".

A: The term “mildew” has been changed into “mould”.

Q3: Could the authors add the number of samples, wood species (botanical name, e.g. Poplus nigra) and surface preparation (sanding, planing) in the material section.

A: The poplar veneer (Chinese White Poplar, uniform material color, 100 mm x 100 mm x 5 mm, 300 pieces, after ordinary mechanical sanding) were supplied by Yihua Lifestyle Technology Co., Ltd., Shantou, China.

Q4: The number of samples/number of replicate measurements are missing.

A: The number of samples is 300. All the experiments were repeated at least four times with an error of less than 5.0%.

Q5: I am missing the method for the formation of "real" nanocrystalline cellulose - the described method refers only to delignification.

A: “CNC” has been changed into “wheat straw powder after lignin removal (WSPALR)”.

Q6: The SEM images are not convincing evidence of the presence of nanocellulose, since the magnification is way too low.

A: The nanocellulose has been deleted.

Q7: Could the authors explain, what purpose the putty serves?

A: The main ingredients of putty are talc powder and water, which are used to fill the vessels of poplar wood.

Q8: Please indicate the dry film thickness of the coating, since this has a major impact on the mechanical properties of the coatings

A: The thickness of the dry waterborne coating was about 40 µm.

Q9: Could the authors explain briefly, how the assessment of the impact test works: how is the impact force varied?

A: In the impact test, the resistance to impact was measured by the QCJ impactor (Tianjin Jingkelian Material Testing Machine Co., Ltd., Tianjin, China). In the impact test, a ball is dropped on the surface of the coating (the maximum distance between the ball and the coating is 50.0 cm, and the weight of the ball is 1.0 kg). At the end of the impact test, the deformation of the coating was observed. If the coating does not crack, the maximum drop height of the ball is the resistance to impact value.

Q10: Could the authors explain, where the Si is coming from?

A: Si, in the form of SiO2, comes from ash in wheat straw powder.

Q11: Surface hardness is one of many performance criteria of coatings and depends on the application. Hence, harder surfaces are not always better than softer ones.

A: The sentence “Surface hardness is one of many performance criteria of coatings but depends on the application. Harder surfaces are not always better than softer ones.” has been added.

Q12: Figure 3 und Figure 4 are not very convincing. Firstly, is it true that a coating with no CNC has a hardness of 1/adhesion of 0/Impact strength of 0 kg cm?.Secondly, are the differences of the datapoints significant?. I am missing some sort of measurement of spread.

A: The lower the adhesion value is, the better the adhesion of the coating is. The adhesion performance of grade 0 is the best. The ordinate unit of hardness in Figure 5 is H, which corresponds to the hard. The resistance to impact of the coatings without WSPALR is about 0, which is very poor. After adding WSPALR and calcined WSPALR, the resistance to impact of the coatings increases to more than 10 kg·cm, which was much higher than that of the coatings without WSPALR and calcined WSPALR. It is true that a coating with no WSPALR has a hardness of 1 H/adhesion of 0 level/resistance to impact of 0 kg·cm. Secondly, all the experiments were repeated at least four times with an error of less than 5.0%.

Round  2

Reviewer 1 Report

No comments. I have read the answers which were satisfying for me.

Reviewer 2 Report

The authors must improve the abstract, to include quantitative (not descriptive) results regarding the properties of the coating

The authors still didn't include the reason for using the calcined cellulose, as the first recommendation suggested 

Pag 2 line 84- what properties are we talking about?

The viscosity of the waterborne wood coating is 20 s.- this must be wrong. Convert it to viscosity units

What is the role of the putty? 

The authors still did not improve the description of the experimental methodology for coatings obtaining, as it was suggested in the first place.

What type of hardness does graph 5 illustrate? Measurement units? Error bars?

How was the adhesion (level) measured? What type of physical quantity is on the Y axis? Error bars? 

The "before" and "after" statements are confusing

The paper does not describe the methodology  for mould resistance testing.

Reviewer 3 Report

The authors have improved the quality of their paper.

However, although the authors claim to have used 300 samples with four replicate mesaurements, only few data points are presented in their study. This makes it very hard to see the significance of their results.

Furthermore, the SEM images in Figure 2 show a very uneven distribution of the WSPALR particles. This makes it hard to believe that this will have an positive effect on the impact resistance of the coatings.